# Coherent noise enables probabilistic sequence replay in spiking neuronal networks

**Younes Bouhadjar**[1,2,3]*, **Dirk J. Wouters**[4], **Markus Diesmann**[1,5], **Tom Tetzlaff**[1]

**1** Institute of Neuroscience and Medicine (INM-6), & Institute for Advanced Simulation (IAS-6), & JARA BRAIN Institute Structure-Function Relationships (INM-10), Jülich Research Centre, Jülich, Germany, **2** Peter Grünberg Institute (PGI-7,10), Jülich Research Centre and JARA, Jülich, Germany, **3** RWTH Aachen University, Aachen, Germany, **4** Institute of Electronic Materials (IWE 2) & JARA-FIT, RWTH Aachen University, Aachen, Germany, **5** Department of Physics, Faculty 1, & Department of Psychiatry, Psychotherapy, and Psychosomatics, Medical School, RWTH Aachen University, Aachen, Germany

* y.bouhadjar@fz-juelich.de

**Data Availability Statement:** The documented workflow and source code necessary to reproduce our findings are provided online at: https://doi.org/10.5281/zenodo.6378376.

## Abstract

Animals rely on different decision strategies when faced with ambiguous or uncertain cues. Depending on the context, decisions may be biased towards events that were most frequently experienced in the past, or be more explorative. A particular type of decision making central to cognition is sequential memory recall in response to ambiguous cues. A previously developed spiking neuronal network implementation of sequence prediction and recall learns complex, high-order sequences in an unsupervised manner by local, biologically inspired plasticity rules. In response to an ambiguous cue, the model deterministically recalls the sequence shown most frequently during training. Here, we present an extension of the model enabling a range of different decision strategies. In this model, explorative behavior is generated by supplying neurons with noise. As the model relies on population encoding, uncorrelated noise averages out, and the recall dynamics remain effectively deterministic. In the presence of locally correlated noise, the averaging effect is avoided without impairing the model performance, and without the need for large noise amplitudes. We investigate two forms of correlated noise occurring in nature: shared synaptic background inputs, and random locking of the stimulus to spatiotemporal oscillations in the network activity. Depending on the noise characteristics, the network adopts various recall strategies. This study thereby provides potential mechanisms explaining how the statistics of learned sequences affect decision making, and how decision strategies can be adjusted after learning.

## Author summary

Humans and other animals often benefit from exploring multiple alternative solutions to a given problem, rather than adhering to a single, global optimum. Such explorative behavior is frequently attributed to noise in the neuronal dynamics. Supplying each neuron or synapse in a neuronal circuit with noise, however, does not necessarily lead to explorative dynamics. If decisions are triggered by the compound activity of ensembles of

**Funding:** This project was funded by the Helmholtz Association Initiative and Networking Fund (project number SO-092, Advanced Computing Architectures) [YB, DJW, MD, TT], and the European Union's Horizon 2020 Framework Programme for Research and Innovation under the Specific Grant Agreement No. 785907 (Human Brain Project SGA2) [YB, MD, TT] and No. 945539 (Human Brain Project SGA3) [YB, MD, TT]. Open access publication funded by the Deutsche Forschungsgemeinschaft (DFG, German Research Foundation, 491111487) [YB, MD, TT]. The funders had no role in study design, data collection and analysis, decision to publish, or preparation of the manuscript.

**Competing interests:** The authors have declared that no competing interests exist.

neurons or synapses, noise averages out, unless it is correlated within these ensembles. As an analogy, consider a particle in a still fluid: despite the constant bombardment by surrounding molecules, a large particle will hardly undergo any Brownian motion, because the momenta of the impinging molecules largely cancel each other. Only if the molecules move in a coherent manner, such as in a turbulent fluid, they can have a substantial influence on the particle's motion. This modeling study exploits this effect to equip a neuronal sequence-processing circuit with explorative behavior by introducing configurable, locally coherent noise. It contributes to an understanding of the neuronal mechanisms underlying different decision strategies in the face of ambiguity, and highlights the role of coherent network activity such as traveling waves during sequential memory recall.

## Introduction

Our brains are constantly processing sequences of events, such as during listening to a song or perceiving the texture of an object. Through repeated exposure to these sequences, we effortlessly learn to predict upcoming events. In many circumstances, we have to make a decision of what elements to recall next in response to a cue. A number of previous modeling studies have proposed spiking neuronal network implementations of sequence learning and replay [1–5]. The spiking temporal-memory (TM) model described in [5] constitutes a biologically more detailed reformulation of the abstract TM algorithm proposed in [6], and provides an energy efficient sequence processing mechanism with high storage capacity by virtue of its sparse activity. It learns complex sequences in an unsupervised, continual manner using biological, local learning rules. After learning, the model successfully predicts upcoming sequence elements in a context dependent manner, and signals the occurrence of non-anticipated stimuli. In contrast to the original TM model in [6], the spiking TM model employs a continuous-time dynamics and predicts that sequences can be successfully learned and processed for a range of sequence speeds with lower and upper bounds determined by electrophysiological parameters such as synaptic and neuronal time constants.

The spiking TM model can be configured into a replay mode where it autonomously recalls learned sequences in response to a cue stimulus. In nature, such cues are often incomplete or ambiguous, and it is not always clear what sequence to recall given the current context. Despite this ambiguity, we usually come to a clear decision on what sequence to recall. A key factor in decision making is reward [7, 8]. In this regard, the optimal decision strategy is the one that maximizes the reward, and is hence referred to as the maximization or exploitation strategy. A number of studies demonstrate that decisions are often made in an apparently suboptimal manner, such as probability matching [9, 10]. In binary choice tasks, for example, where the probability of payoff is higher for one of the two possible choices, it appears most reasonable to always decide for this high-payoff option. Instead, however, humans and other animals often decide for each of the two choices with a probability that approximately matches the payoff probability. While this behavior appears unreasonable at first glance, it may in fact be optimal when taking into account previous (pre-experiment) experiences, such as prior knowledge of changing reward contingencies. In cases where the reward probability or amplitudes change over time, a more explorative behavior is beneficial [7, 11]. Previous studies suggest that decisions are not only determined by rewards, but also by the frequency of previously experienced input patterns [12, 13]. Accordingly, suboptimal decision strategies may at least partly arise as a consequence of this additional influence of occurrence frequencies.

A number of previous studies propose neuronal network models of decision making in the face of ambiguous or incomplete stimuli. The majority of these models employ some form of intrinsic stochastic dynamics or uncorrelated noise to generate explorative behavior [14–18]. Noise has been introduced in the form of random or non-task-related synaptic background inputs [18], or in the form of synaptic stochasticity [17]. An alternative solution is proposed in [16, 19], where the "noise" is generated by the complex but deterministic dynamics of the functional network itself, without any additional sources of stochasticity. In most models, the noise targeting different neurons or synapses is effectively uncorrelated. Supplying each element in a neuronal circuit with uncorrelated noise, however, does not necessarily lead to explorative dynamics: state variables arising from superpositions of many noisy, uncorrelated components become effectively deterministic as a result of noise averaging [19]. The total input current of a neuron generated from superpositions of many synaptic inputs, for example, is hardly affected by the variability in the individual synaptic responses. Similarly, in models where individual states are encoded by the activity of neuronal subpopulations [15], the state representations become quasi deterministic if the single-neuron noise components are uncorrelated. Compensating this noise averaging effect by increasing the noise amplitude appears to be an obvious strategy, but may be hard to realize by the biological system.

An alternative, natural solution to the noise-averaging problem is to employ locally correlated noise. In biological neuronal networks, coherent noise may arise by different mechanisms: neighboring neurons typically receive inputs from partly overlapping presynaptic neuron populations. The synaptic input currents to these neurons are therefore correlated. In the literature, this type of correlation, which results from the anatomy of neurons and neuronal circuits, is referred to as shared-input correlation [20, 21]. A second type of correlation in synaptic input currents arises from correlations in the presynaptic spiking activity [22–24]. These dynamical correlations occur during stationary network states, or can be generated by different types of nonstationary activities, such as global oscillations in the population activity [25, 26] or traveling waves of activity propagating across the neuronal tissue [27–30].

This study addresses the problem of sequential decision making in the face of ambiguity and the role of coherent noise in shaping decision strategies. We investigate how the spiking TM model in [5] recalls sequences in response to ambiguous cues in the presence of locally coherent noise, to what extent noise averaging can be overcome by increasing the noise amplitude, and how different recall strategies can be achieved by adjusting the noise characteristics. We further explore whether shared synaptic input and random stimulus locking to spatiotemporal oscillations can serve as appropriate, natural sources of coherent noise. In Materials and methods, we provide a detailed description of the task and the network model.

## Results

### A spiking neural network recalls sequences in response to ambiguous cues

In this section, we provide a brief overview of the model and the task, illustrate how the network learns overlapping sequences occurring with different frequencies during the training, and show how these occurrence frequencies are encoded in the network. We then study the network responses to ambiguous cues and the influence of the occurrence frequencies on the recall behavior in the absence or presence of noise.

Similar to [5], the model consists of a randomly and sparsely connected network of $N_{\mathrm{E}}$ excitatory neurons (population $\mathcal{E}$) and a single inhibitory neuron (Fig 1A). Each excitatory neuron receives $K_{\mathrm{EE}}$ excitatory inputs from other randomly chosen neurons in $\mathcal{E}$. Excitatory neurons are subdivided into $M$ subpopulations, each containing neurons with identical stimulus preference: in the absence of any additional connections, all neurons in a given

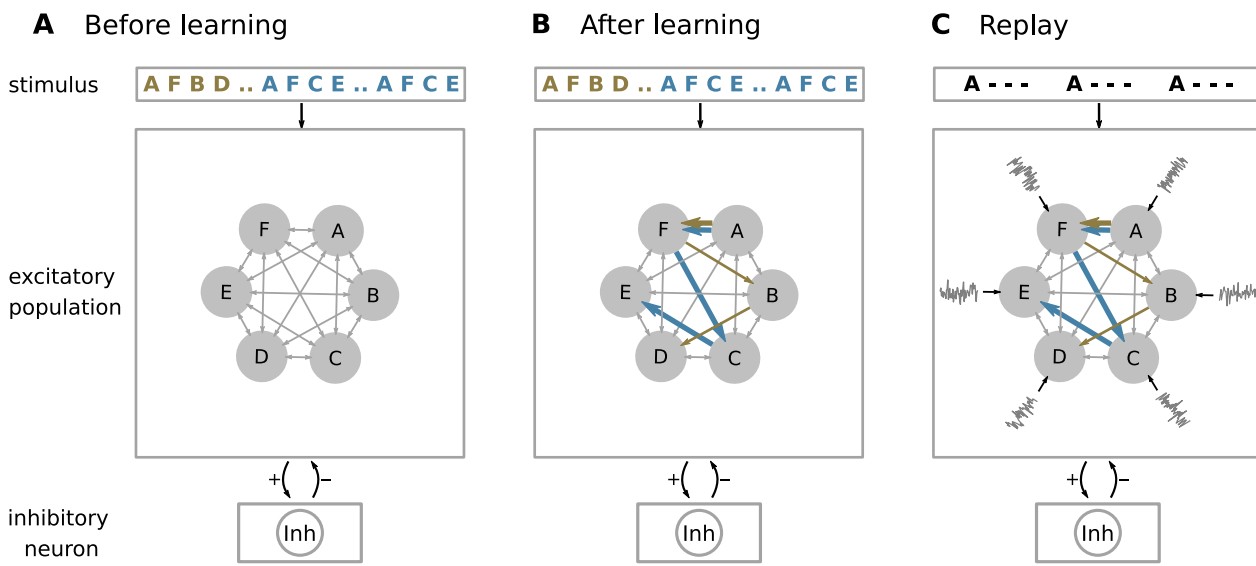

**Fig 1. Network structure. A)** The architecture constitutes a recurrent network of subpopulations of excitatory neurons (filled gray circles) and a single inhibitory neuron (Inh). Each excitatory subpopulation contains neurons with identical stimulus preferences. Excitatory neurons are stimulated by external sources providing sequence-element specific inputs "A","F", "B", etc. Connections between and within the excitatory subpopulations are random and sparse. The inhibitory neuron is recurrently connected to all excitatory neurons. In the depicted example, the network is repetitively presented with two sequences {A,F,B,D} (brown) and {A,F,C,E} (blue) during learning. The sequence {A,F,C,E} occurs twice as often as {A,F,B,D}. **B)** During learning, the network forms sequence specific subnetworks (blue and brown arrows representing {A,F,B,D} and {A, F,C,E}, respectively) as a result of the synaptic plasticity dynamics. The connections between subpopulations representing the sequence shown more often are stronger (thick arrows). **C)** The network can be configured into a replay mode by increasing the neuronal excitability. During the replay mode, the network is presented with a cue stimulus representing the first sequence element "A". In addition, the excitatory subpopulations receive input from distinct sources of background noise (gray traces) which is not present during learning. In the replay mode, the synaptic plasticity is switched off.

subpopulation fire a spike upon the presentation of a specific sequence element. The inhibitory neuron is recurrently connected to the excitatory neurons. In contrast to [5] where each excitatory subpopulation is equipped with its own inhibitory neuron, we here use a single inhibitory neuron to implement a winner-take-all (WTA) competition between the subpopulations of excitatory neurons. At the same time, the inhibitory neuron mediates the competition between neurons within subpopulations and thereby leads to sparse activity and context sensitivity, as described in [5] and below. The network is driven by external inputs, each representing a specific sequence element ("A", "B", . . .), and feeds all neurons in the subpopulation $\mathcal{M}_k$ that have the same stimulus preference. Neurons are modeled as point neurons with the membrane potential evolving according to the leaky integrate-and-fire dynamics [31]. The total synaptic input current of excitatory neurons is composed of currents in distal dendritic branches, inhibitory currents, and currents from external sources, see Eq (5). The inhibitory neuron receives only inputs from excitatory neurons. The dynamics of dendritic currents include a nonlinearity describing the generation of dendritic action potentials (dAPs), see Eq (10). Synapses between excitatory neurons are plastic and subject to spike-timing-dependent plasticity and homeostatic control. Details on the network model are given in Materials and methods.

During the learning, the network is exposed to repeated presentations of $S$ sequences $s_1$, . . ., $s_S$, such that each sequence $s_i$ occurs with a specific frequency $p_i$ (for details on the learning protocol, see Materials and methods). For illustration, we focus here on a simple set of two sequences {A,F,B,D} and {A,F,C,E}, where the first sequence is shown with a relative frequency $p_1 = p$ and the second with $p_2 = 1 - p$ (e.g., $p = 0.2$ in Fig 2A). In the following, we refer to {A,F,

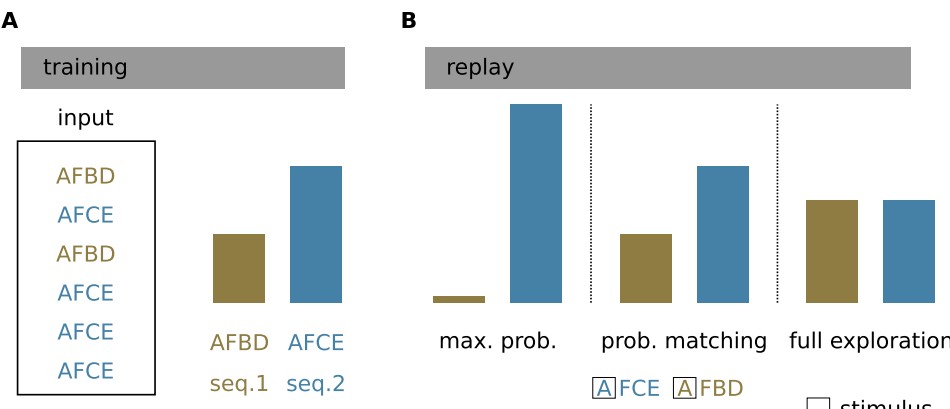

**Fig 2. Task. A)** During learning, the model is exposed to two (or more) competing sequences with different frequencies. Here, sequence 1 ({A,F,B,D}; brown) is shown twice as often as sequence 2 ({A,F,C,E}; blue). The respective normalized training frequencies $p_1 = 1/3$ and $p_2 = 2/3$ are depicted by the histogram. **B)** During replay, the network autonomously recalls the sequences in response to an ambiguous cue (first sequence element "A"; open black squares) according to different strategies. Maximum probability (max-prob): only the sequence with the highest training frequency is replayed. Probability matching (prob. matching): the replay frequency of a sequence matches its training frequency. Full exploration: all sequences are randomly replayed with the same frequency, irrespective of the training frequency. Histograms represent the replay frequencies $f_{\{s_1\}}$ and $f_{\{s_2\}}$, respectively.

B,D} as sequence 1 and to {A,F,C,E} as sequence 2. Before learning, presenting a sequence element causes all neurons in the respective subpopulation to fire. During the learning process, the repetitive sequential presentation of sequence elements increases the strength of connections between the corresponding subpopulations to a point where the activation of a certain subpopulation by an external input generates dAPs in a specific subset of neurons in the subpopulation representing the subsequent element. The generation of the dAPs results in a long-lasting depolarization ($\sim 50 - 500$ ms) of the soma. We refer to neurons that generate a dAP as predictive neurons. When receiving an external input, predictive neurons fire earlier as compared to non-predictive neurons. If a group of at least $\rho$ neurons are predictive within a certain subpopulation, their advanced spikes initiate a fast and strong inhibitory feedback to all excitatory neurons, ultimately suppressing the firing of non predictive neurons. After learning, the model develops specific subnetworks representing the learned sequences (Fig 1B), such that the presentation of a sequence element leads to a context dependent prediction of the subsequent element [5]. As a result of Hebbian learning, the synaptic weights in the subnetwork corresponding to the most frequent sequence during learning are on average stronger than those for the less frequent sequence (Figs 1B, 3A and 4A). In the prediction mode, this asymmetry in synaptic weights plays no role. For ambiguous stimuli, all potential outcomes are predicted, i.e., the network predicts both "C" and "B" simultaneously in response to stimuli "A" and "F", irrespective of the training frequencies.

The model can be configured into a replay mode, where the network autonomously replays learned sequences in response to a cue stimulus. This is achieved by changing the excitability of the neurons such that the activation of a dAP alone can cause the neurons to fire [5]. In addition, the synaptic plasticity is disabled during replay to preserve the encoding of the training frequencies in the synaptic weights (Fig 4A; see also Discussion). In the replay mode, we present ambiguous cues and study whether the network can replay sequences following different strategies (Fig 2B). We refer to the "maximum probability" strategy (Fig 2B, left) as the case where the network exclusively replays the sequence with the highest occurrence frequency during training. When adopting the "probability matching" strategy, the network replays

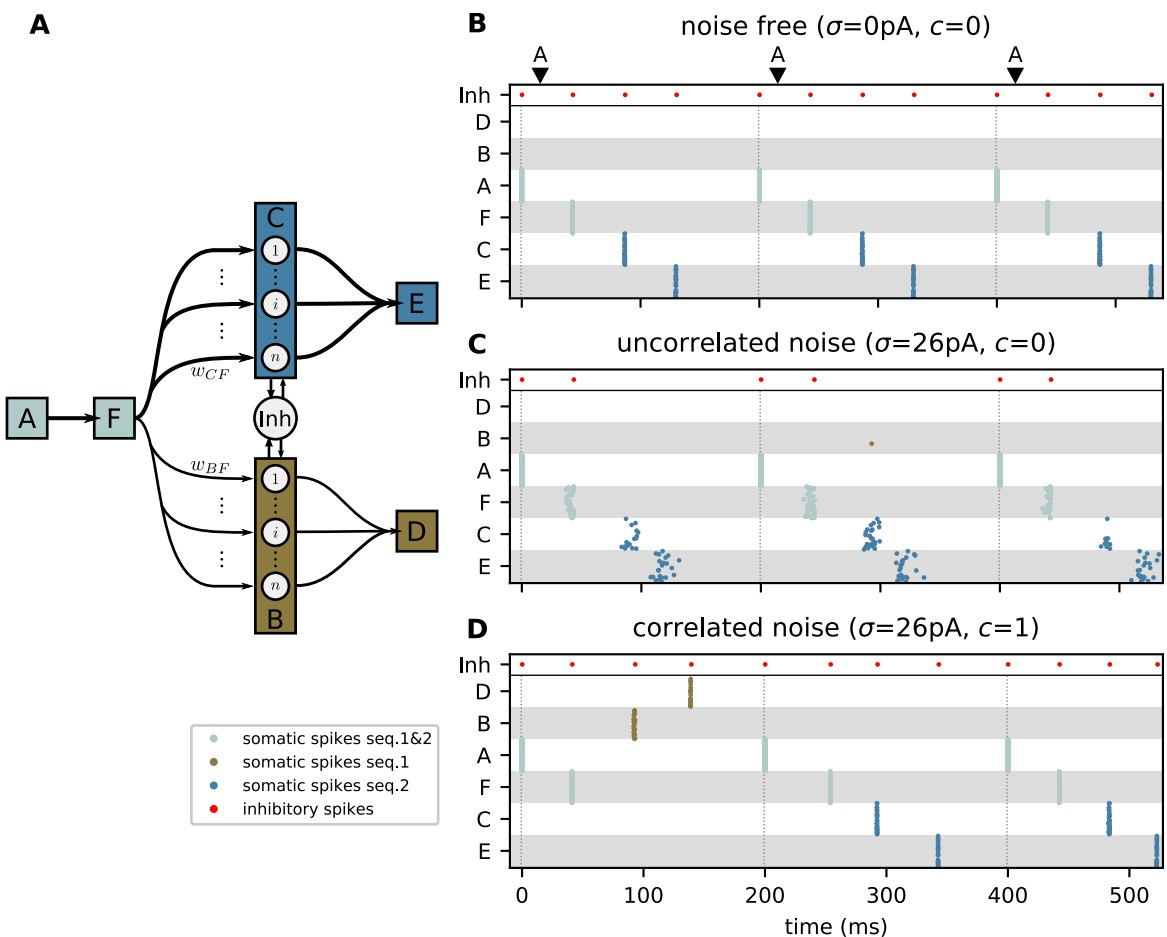

**Fig 3. Correlated noise enhances exploratory behavior. A)** Sketch of subpopulations of excitatory neurons (boxes) representing the elements of the two sequences {A,F,C,E} (seq. 2) and {A,F,B,D} (seq. 1). The subpopulations "C" and "B" are unfolded showing their respective neurons. The arrows depict the connections after learning the task shown in Fig 2A. The line thickness represents the population averaged synaptic weight. The presentation of the character "A" constitutes an ambiguous cue during replay. The inhibitory neuron (Inh) mediates competition between subpopulations through the winner-take-all (WTA) mechanism. **B,C,D)** Spiking activity in the subpopulations depicted in panel A in response to three repetitions of the ambiguous cue "A" (black triangles at the top and vertical dotted lines) for three different noise configurations $\sigma = 0$ pA, $c = 0$ (B), $\sigma = 26$ pA, $c = 0$ (C), and $\sigma = 26$ pA, $c = 1$ (D). Brown, blue, and silver dots mark somatic spikes of excitatory neurons corresponding to sequence 1, sequence 2, and both, respectively. For clarity, only the sparse subsets of active neurons in each population are shown. Red dots mark spikes of the inhibitory neuron. Panels C and D depict the representative recall behavior. See Fig 4 for a detailed statistics across trials and network realizations. See Table 9 for model parameters.

sequences with a frequency that matches the training frequency (Fig 2B, middle). The "full exploration" strategy refers to the case where all sequences are randomly replayed with the same frequency, irrespective of the training frequency (Fig 2B, right). In Fig 3, we illustrate the network's decision behavior by presenting the ambiguous cue stimulus "A" three times. In the absence of noise, the network adopts the maximum probability strategy (Fig 3B): as a result of the higher weights between the neurons representing the more frequent sequence, the dAPs are activated earlier in these neurons, which advances their somatic firing times with respect to the neurons representing the less frequent sequence. This advanced response time quickly activates the inhibitory neuron, which suppresses the activity of the other neurons.

To assess the replay performance, we present the ambiguous cue "A" for $N_t$ trials and examine the replay frequencies $f_{\{s_1\}}$ and $f_{\{s_2\}}$ of the two sequences $s_1 = \{A,F,B,D\}$ and $s_2 = \{A,F,C,E\}$ as a function of their relative occurrence frequencies $p_i$ during training. We define the

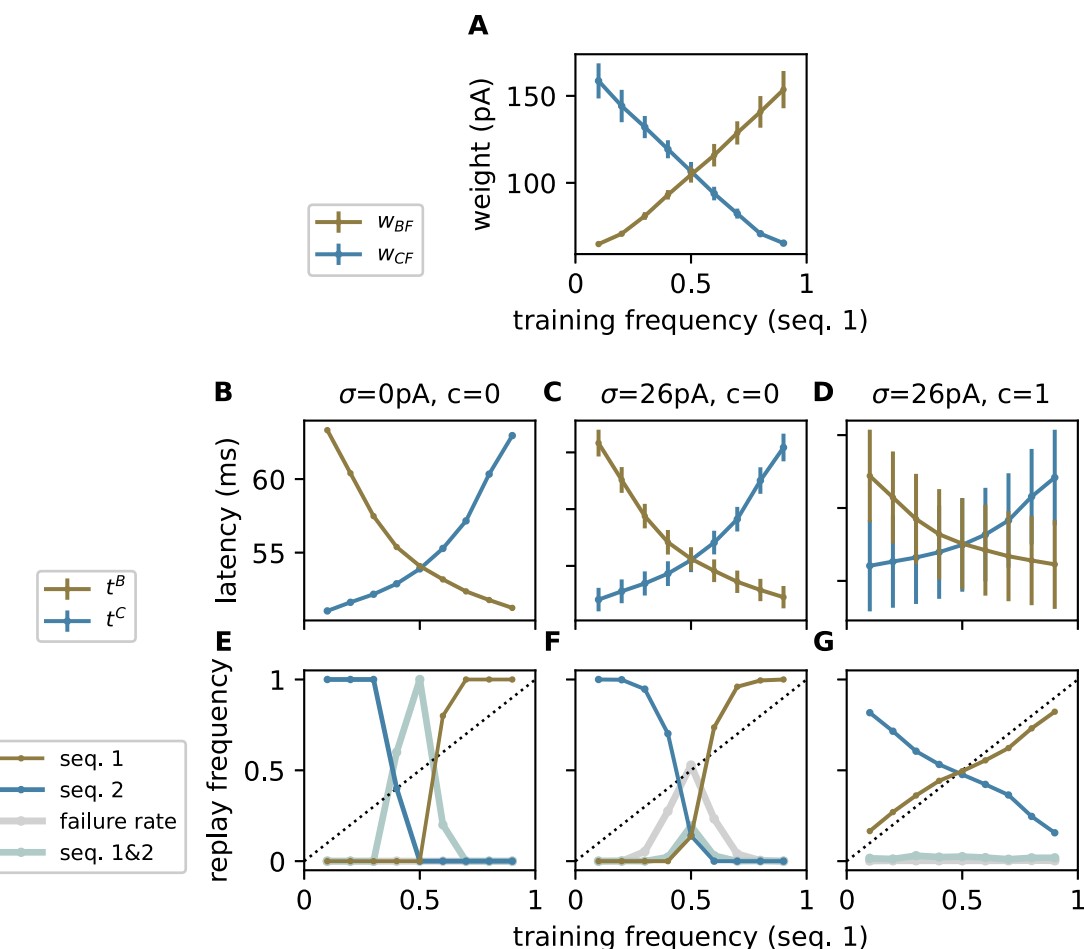

**Fig 4. Uncorrelated noise averages out in population based encoding.** Dependence of **A)** the compound weights (PSC amplitudes) $w_{BF}$ (brown) and $w_{CF}$ (blue; see Fig 3A), **B–D)** the population averaged response latencies $t^B$ and $t^C$ (subpopulation averaged time of first spike after the cue "A"; see Eq (1) for subpopulations "B" (brown) and "C" (blue), and **E–G)** the relative replay frequencies $f_{\{s_1\}}$ and $f_{\{s_2\}}$ of sequences 1 (brown) and 2 (blue), the failure rate $f_\emptyset$ (gray) and the joint probability $f_{\{s_1,s_2\}}$ of replaying both sequences (silver) on the training frequency $p_1 = p$ of sequence 1. Note that the inhibition is disabled when measuring the latencies to ensure that both competing populations "B" and "C" elicit spikes. Panels B–G depict results for three different noise configurations $\sigma = 0$ pA, $c = 0$ (B,E), $\sigma = 26$ pA, $c = 0$ (C,F), and $\sigma = 26$ pA, $c = 1$ (D,G). In panel A, circles and error bars depict the mean and the standard deviation across different network realizations. In panels B–D, circles and error bars represent the mean and the standard deviation across $N_t = 151$ trials (cue repetitions), averaged across 5 different network realizations. In panels E–G, circles represent the mean across $N_t = 151$ trials, averaged across 5 different network realizations. See Table 9 for remaining parameters. Same task as described in Fig 2.

sequences {A,F,B,D} or {A,F,C,E} to be successfully replayed if more than $0.5\rho = 10$ neurons in the last subpopulations "E" or "D" have fired, respectively (for details on the assessment of the replay statistics, see Materials and methods). In the absence of noise, the network replays only the sequence with the highest training frequency $p$ (Fig 4E). To understand this behavior, we inspect the response latencies $t^{B/C}$ of the subpopulations "B" and "C" as a function of the training frequencies (Fig 4B). Here, the response latency

$$t^x = \frac{1}{\rho} \sum_{i \in \mathcal{X}}^{\rho} t_i \tag{1}$$

of the subpopulation $\mathcal{X}$ representing sequence element $x \in \{B,C\}$ corresponds to the

population average of the single-neuron response latencies $t_i$ (time of first spike after the cue) for each active neuron $i \in \mathcal{X}$ in this subpopulation. Averaged across trials, the response latency is smaller for the subpopulation participating in the sequence with the higher frequency. The response latencies $t^B$ and $t^C$ decrease with increasing the respective training frequencies. In the absence of noise, the distribution of the response latencies $t^{B/C}$ across trials is very narrow ([Fig 4B]). Consequently, neurons representing the most frequent sequence fire earlier in all trials. For training frequencies between 0.4 and 0.6, the difference between $t^B$ and $t^C$ in some network realizations is small compared to the response latency of the WTA circuit. Hence, both sequences are occasionally replayed simultaneously ([Fig 4E]).

To foster exploratory behavior, i.e., to enable occasional replay of the low-frequency sequence, we equip the excitatory neurons with background noise. For simplicity, this background noise is added only during replay, but not during the learning (see [Discussion]). In this work, we investigate two different forms of background noise. Here, we first consider noise provided in the form of stationary synaptic background input (see below for an alternative form of noise). To this end, each subpopulation of excitatory neurons receives input from its private pool of independent excitatory and inhibitory Poissonian spike sources ([Fig 1C]). The background noise is parameterized by the noise amplitude $\sigma$ (standard deviation of the synaptic input current arising from these background inputs) and the noise correlation $c$ (see [Fig 1C] and [Materials and methods]). Inputs to neurons of the same subpopulation are correlated by an extent parameterized by $c$. Neurons in different subpopulations receive uncorrelated inputs. The noise amplitude $\sigma$ is chosen such that the subthreshold membrane potentials of the excitatory neurons are fluctuating without eliciting additional spikes. As a consequence, the distributions of response latencies $t^{B/C}$ across trials may be broadened and partly overlap ([Fig 4C and 4D]). As we will show in the following, the network can adopt different replay strategies ([Fig 2B]) depending on the amount of this overlap. Note that noise is injected only during replay, but not during learning. During training, the weak noise employed here hardly affects the network behavior as the external inputs (stimulus) are strong and lead to a reliable, immediate responses.

With uncorrelated noise ($c = 0$), the replay behavior remains effectively non-explorative, i.e., only the high-frequency sequence is replayed in response to the cue ([Fig 3C]). This is explained by the fact that each sequence element is represented by a subset of $\rho$ neurons, or in other words, that the response latency $t^x$ in [Eq (1)] is a population averaged quantity. Its across-trial variance

$$v^x = \mathrm{Var}(t^x) = \frac{1}{\rho} v_\mathrm{s} + \frac{\rho - 1}{\rho} c_\mathrm{s} v_\mathrm{s} \tag{2}$$

is determined by the population size $\rho$, the population averaged spike-time variance $v_\mathrm{s} = \frac{1}{\rho} \sum_i^\rho \mathrm{Var}(t_i)$, and the population averaged spike-time correlation coefficient $c_\mathrm{s} = \frac{1}{\rho(\rho-1)v_\mathrm{s}} \sum_i^\rho \sum_{j \neq i}^\rho \mathrm{Cov}(t_i, t_j)$, with $\mathrm{Cov}(t_i, t_j)$ denoting the spike-time covariance for two neurons $i$ and $j$. Here, we use the subscript "s" to indicate that $v_\mathrm{s}$ and $c_\mathrm{s}$ refer to the (co-)variability in the (first) "spike" times. The spike-time statistics $v_\mathrm{s}$ and $c_\mathrm{s}$ depend on the input noise statistics $\sigma$ and $c$ in a unique and monotonous manner [32, 33]. In the absence of correlations ($c = c_\mathrm{s} = 0$), the across-trial variance $v$ of $t^x$ vanishes for large population sizes $\rho$. For finite population sizes, $v$ is non-zero but small ([Fig 4C]). The effect of the synaptic background noise on the variability of response latencies largely averages out. Hence, the average advance in the response of the population representing the high-frequency sequence cannot be overcome by noise; the network typically replays only the sequence with the higher occurrence frequency during training ([Fig 4F]). For small differences in the training frequencies ($p \approx 0.5$), the

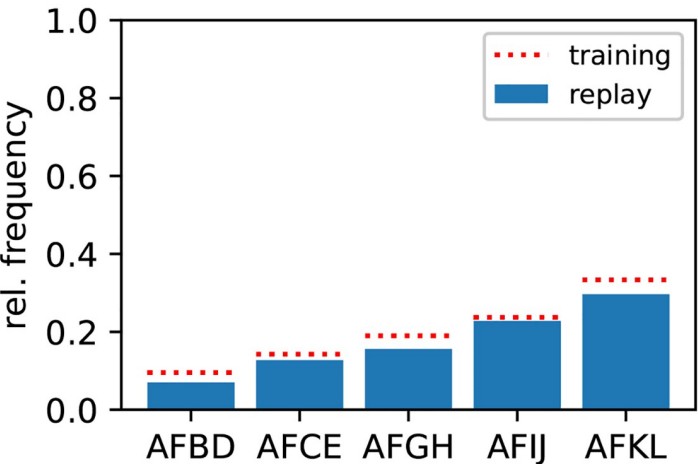

**Fig 5. Multiple competing sequences are learned and replayed according to their occurrence frequencies (probability matching).** During learning, five competing, partly overlapping sequences $s_1 = \{A,F,B,D\}$, $s_2 = \{A,F,C,E\}$, $s_3 = \{A,F,G,H\}$, $s_4 = \{A,F,I,J\}$, and $s_5 = \{A,F,K,L\}$ are repetitively presented with relative training frequencies $p_1 = 0.1$, $p_2 = 0.14$, $p_3 = 0.2$, $p_4 = 0.23$, $p_5 = 0.33$, respectively (dotted red lines). After learning, the network autonomously replays the learned sequences in response to the ambiguous cue "A" with frequencies $f_{\{s_1\}}, f_{\{s_2\}}, \ldots, f_{\{s_5\}}$ depicted by the blue bars. Parameters: $\sigma = 12$ pA, $c = 1$, $\tau_\mathrm{h} = 4620$ ms, $z^* = 21$, $N_\mathrm{e} = 101$, $M = 12$. See Table 9 for remaining parameters.

network occasionally fails to replay any sequence or replays both sequences. The mechanism underlying this behavior is explained below.

Noise averaging is efficiently avoided by introducing noise correlations. For perfectly correlated noise and, hence, perfectly synchronous spike responses ($c = c_\mathrm{s} = 1$), the across-trial variance $v$ of the response latency $t$ is identical to the across-trial variance $v_\mathrm{s}$ of the individual spike responses, i.e., $v = v_\mathrm{s}$, irrespective of the population size $\rho$; see Eq (2). For smaller but non-zero spike correlations ($0 < c_\mathrm{s} < 1$), the latency variance $v$ is reduced but doesn't vanish as $\rho$ becomes large. Hence, in the presence of correlated noise, the across-trial response latency distributions for two competing populations have a finite width and may overlap (Fig 4D), thereby permitting an occasional replay of the sequence observed less often during training (Figs 3D and 4G and S6 Fig). Replay, therefore, becomes more exploratory, such that the occurrence frequencies during training are gradually mapped to the frequencies of sequence replay. With an appropriate choice of the noise amplitude and correlation, even an almost perfect match between training and replay frequencies can be achieved (probability matching; Fig 4G). For a training frequency $p = 0.2$, the replay frequency matches $p$ already after about 20 training episodes (S5 Fig).

The results presented so far can be extended towards more than two competing sequences. As a demonstration, we train the network using five sequences $\{A,F,B,D\}$, $\{A,F,C,E\}$, $\{A,F,G,H\}$, $\{A,F,I,J\}$, and $\{A,F,K,L\}$ presented with different relative frequencies. By adjusting the noise amplitude $\sigma$ and correlation $c$, the replay frequencies can approximate the training frequencies (Fig 5).

## Noise averaging cannot be overcome by increasing noise amplitude

For subpopulations of finite size $\rho$, the variance $v$ of the response latency $t$ remains finite, and can be increased by scaling up the variance of the noise, even without correlation; see Eq (2). However, this solution cannot be applied to network models where a decision is mediated by a fast WTA circuit. In the presence of uncorrelated noise with high amplitude, the spikes from all neurons, in all competing subpopulations, are similarly dispersed. A large dispersion in spike times prohibits a fast and reliable activation of inhibition by one of the competing subpopulations. The WTA mechanism, therefore, fails at selecting a unique sequence. Consequently, both sequences

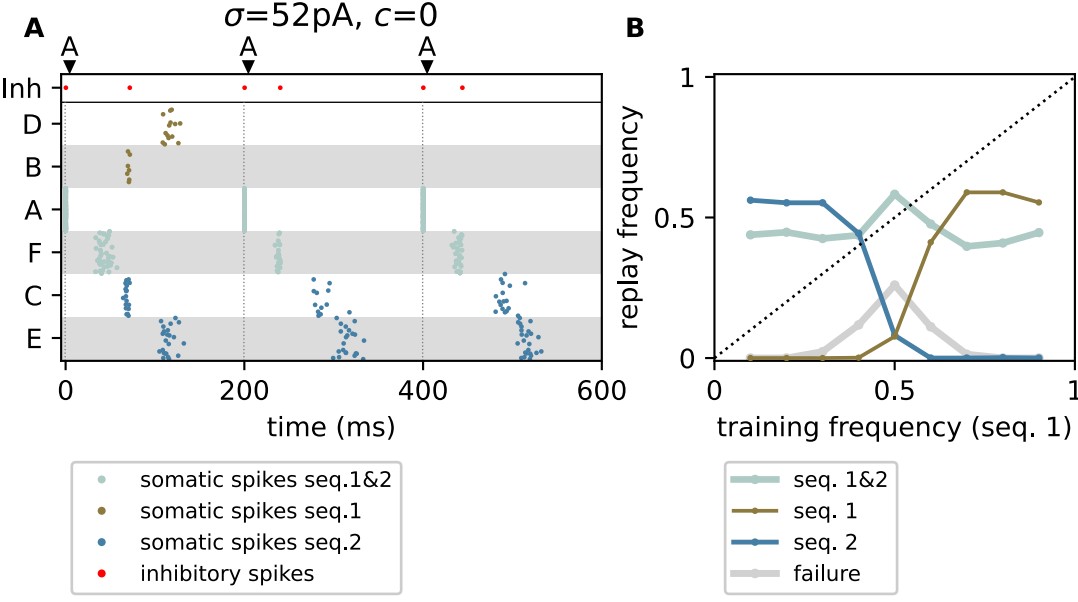

**Fig 6. Winner-take-all mechanism fails when increasing the amplitude of the uncorrelated noise. A)** Brown, blue, and silver dots mark somatic spikes of excitatory neurons belonging to sequence {A,F,B,D} (seq. 1), sequence {A,F,C,E} (seq. 2), or both, respectively. Red dots mark spikes from the inhibitory neuron. Each trial is initiated by stimulating the first element in the sequence ("A", see dark arrows and vertical dashed lines). During training, the sequences 1 and 2 are shown with relative frequencies $p_1 = 0.3$ and $p_2 = 0.7$, respectively. **B)** Dependence of the relative replay frequencies $f_{\{s_1\}}$ and $f_{\{s_2\}}$ of sequence 1 (brown) and sequence 2 (blue), the failure rate $f_\emptyset$ (gray), and the joint probability $f_{\{s_1,s_2\}}$ of replaying both sequences (silver) on the relative training frequency $p_1 = p$ of sequence 1. Circles represent the mean across $N_t = 151$ trials averaged across 5 network realizations. Parameters: $\sigma = 52$ pA and $c = 0$. See Table 9 for the remaining parameters. Same task as described in Fig 2.

run through in most of the trials (Fig 6A). An additional problem of the uncorrelated noise is that it impairs the propagation of the activity across the subpopulations of neurons. As our model relies on the propagation of synchronously firing neurons, the spike time dispersion resulting from the uncorrelated noise bears the risk that the spikes generated may be too dispersed to trigger dAPs in the next subpopulation (Fig 6). As a result of these two problems, more explorative behavior cannot be achieved by increasing the amplitude of uncorrelated noise. Instead, the probability of simultaneous replay (no decision) and the failure rate increase (Fig 6B).

Noise correlations lead to more synchronous responses, thereby reducing the overlap between the within-trial latency distributions of the two competing populations "B" and "C" (Fig 3D). In each trial, the WTA dynamics is therefore triggered by just one of the two populations, rather than by both. Further, synchronous firing leads to a more robust activation of the subsequent subpopulation, and hence, a more robust replay. Hence, noise correlations help not only in generating more explorative behavior, but also in reducing replay failures and the chance of simultaneous activation of competing sequences (Fig 4G).

## Noise amplitude and level of correlation control replay strategy

Psychophysics studies show that humans and other animals can flexibly change their decision strategies in the face of uncertainty or ambiguity [7, 11]. In the context of the model proposed here, this behavior is reproduced by adjusting the characteristics of the noise: by varying the noise amplitude, the model can be tuned to adopt a maximum-probability (Fig 7A), a probability-matching (Fig 7B), or an even more exploratory replay strategy (Fig 7C), provided the noise correlations are sufficiently strong. Similarly, it may be possible to change the replay behaviors by varying the noise correlation level (S1 Fig), if some of the model parameters are

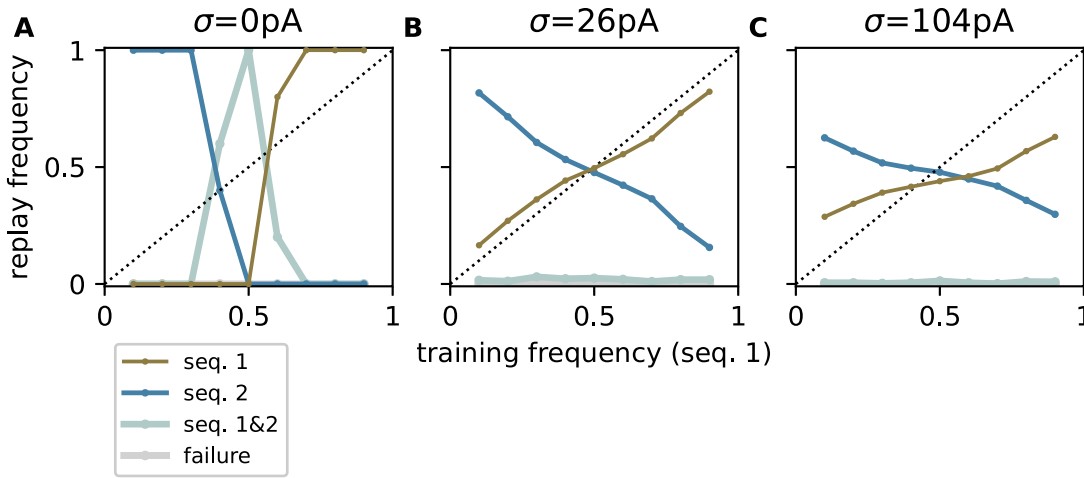

**Fig 7. Different replay strategies achieved by increasing the noise amplitude.** Dependence of the relative replay frequencies $f_{\{s_1\}}$ and $f_{\{s_2\}}$ of sequence 1 (brown) and sequence 2 (blue), the failure rate $f_\emptyset$ (gray) and the joint probability $f_{\{s_1,s_2\}}$ of replaying both sequences (silver) on the relative training frequency $p_1 = p$ of sequence 1 for different noise amplitudes $\sigma = 0$ pA (**A**), $\sigma = 26$ pA (**B**), and $\sigma = 104$ pA (**C**) with correlation coefficient $c = 1$. Circles represent the mean across $N_t = 151$ trials, averaged across 5 different network realizations. See Table 9 for remaining parameters. Same task as described in Fig 2.

adjusted during replay, especially to ensure a robust activity propagation of the less frequent sequence (e.g., by decreasing $J_{EI}$). In nature, a modulation of the noise amplitude is achieved by changing the firing rate of the presynaptic neurons providing the background noise, or the excitability of the target neurons via neuromodulatory [34] or attention signals [35].

So far, we discussed shared stationary presynaptic input as a potential source of correlated noise occurring in nature. Shared input correlations resulting from the anatomy of cortical circuits are low [36–39]. To generate explorative replay behavior in the context of our model, however, the level of noise correlation needs to be substantial ($c \sim 1$). In the following section, we therefore propose an alternative form of noise, where high correlations arise from the network dynamics.

## Random stimulus locking to spatiotemporal oscillations as natural form of noise

In vivo cortical activity is rarely stationary. Usually, it is characterized by substantial temporal and spatial fluctuations, often occurring in the form of transient spatiotemporal oscillations, i.e., cortical waves [27, 40–42]. In the presence of traveling cortical waves, nearby neurons share the same oscillation phase, whereas distant neurons experience different phases (Fig 8). At the time of stimulus arrival, neurons in the up phase are more excitable and tend to fire earlier than neurons in a down phase. Cortical waves can be locked to external stimuli or events such as saccades [43], but they also occur spontaneously without locking to external cues [44]. Here, we exploit this finding and assume that the cue onset times are random with respect to the oscillation phase, thereby introducing a locally coherent form of trial-to-trial variability during replay.

To investigate the effect of this type of variability on the replay performance, we first train the network in the absence of any background input using the same two-sequence task and training setup discussed in earlier sections. During replay, we inject an oscillating background current with amplitude $a$ and frequency $f$ into all excitatory neurons (see Materials and methods). Neurons within a given subpopulation share the same oscillation phase. Phases for different subpopulations are randomly drawn from a uniform distribution between 0 and $2\pi$. The

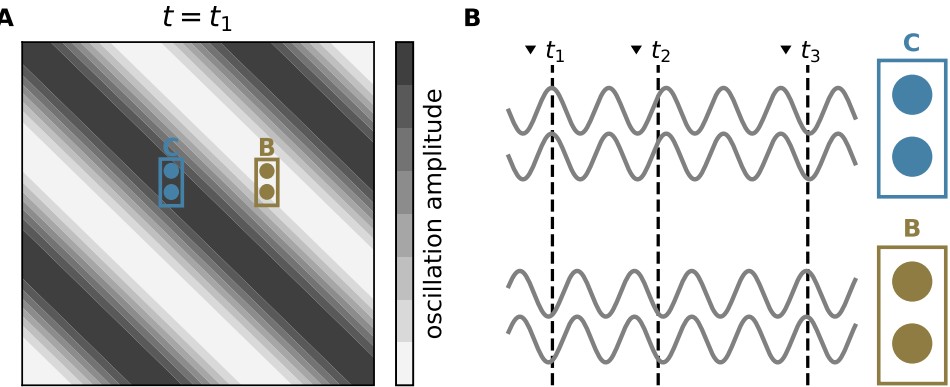

**Fig 8. Random locking of stimulus to global oscillations as a form of noise. A)** Snapshot of a wave of activity traveling across a cortical region at time $t_1$ of the 1st stimulus onset. Grayscale depicts wave amplitudes in different regions. Brown and blue rectangles mark populations of neurons with stimulus preferences "B" and "C", respectively. **B)** Background inputs to neurons in populations "B" and "C" at different times. Background inputs to each population "B" and "C" at different times. Background inputs to neurons within each population are in phase due to their spatial proximity. Background inputs to different populations are phase shifted. Arrows on the top depict stimulus onset times. The times $t_1, t_2, \ldots$ indicate input arrival to populations "B" and "C" (dashed vertical lines are random, not locked to the background activity).

replay performance of the network is assessed by monitoring the network responses to repetitive presentations of an external cue "A" with random, uniformly distributed inter-cue intervals $\Delta T_{\text{cue}} \sim \mathcal{U}(u_{\min}, u_{\max})$. The analysis is repeated for a range of training frequencies $p$, oscillation amplitudes $a$, and frequencies $f$.

Depending on the choice of the oscillation amplitude $a$ and frequency $f$, the network replicates different replay strategies (Fig 9). For low-amplitude oscillations, the model replays only the sequence with the higher training frequency (max-prob). With increasing oscillation amplitude, it becomes more explorative and occasionally replays the less frequent sequence. By adjusting the oscillation amplitude, the replay frequency can be closely matched to the training frequency. This behavior of the model is observed for a range of physiological frequency bands such as alpha ($\sim 10$ Hz), beta ($\sim 30$ Hz), and gamma ($\sim 70$ Hz) [45, 46]. Higher oscillation frequencies are less effective due to the low-pass characteristics of neuronal membranes and synapses. Consequently, increasing the oscillation frequency leads to a more reliable replay of the most frequent sequence. For slow oscillations with long periods that are large compared to the average inter-cue interval, the network responses in subsequent trials are more correlated. For sufficiently many trials, however, the network can still explore different solutions.

To conclude: cortical waves in a range of physiological frequencies represent a form of highly fluctuating and locally correlated background activity. The absence of a systematic stimulus locking to this activity constitutes a natural source of randomness that does not average out and is hence well suited to generate robust exploratory behavior. The degree of exploratoriness, i.e., the decision strategy, can be adjusted in a biologically plausible manner by controlling the wave amplitude or frequency.

## Discussion

This work proposes a spiking neuronal network model performing probabilistic sequential memory recall in response to ambiguous cues. Explorative recall is achieved by providing the network with locally coherent noise. We explore two forms of this noise, implemented either in the form of shared synaptic input or a random stimulus locking to global spatiotemporal oscillations in the neuronal activity. The model explains the emergence of different recall

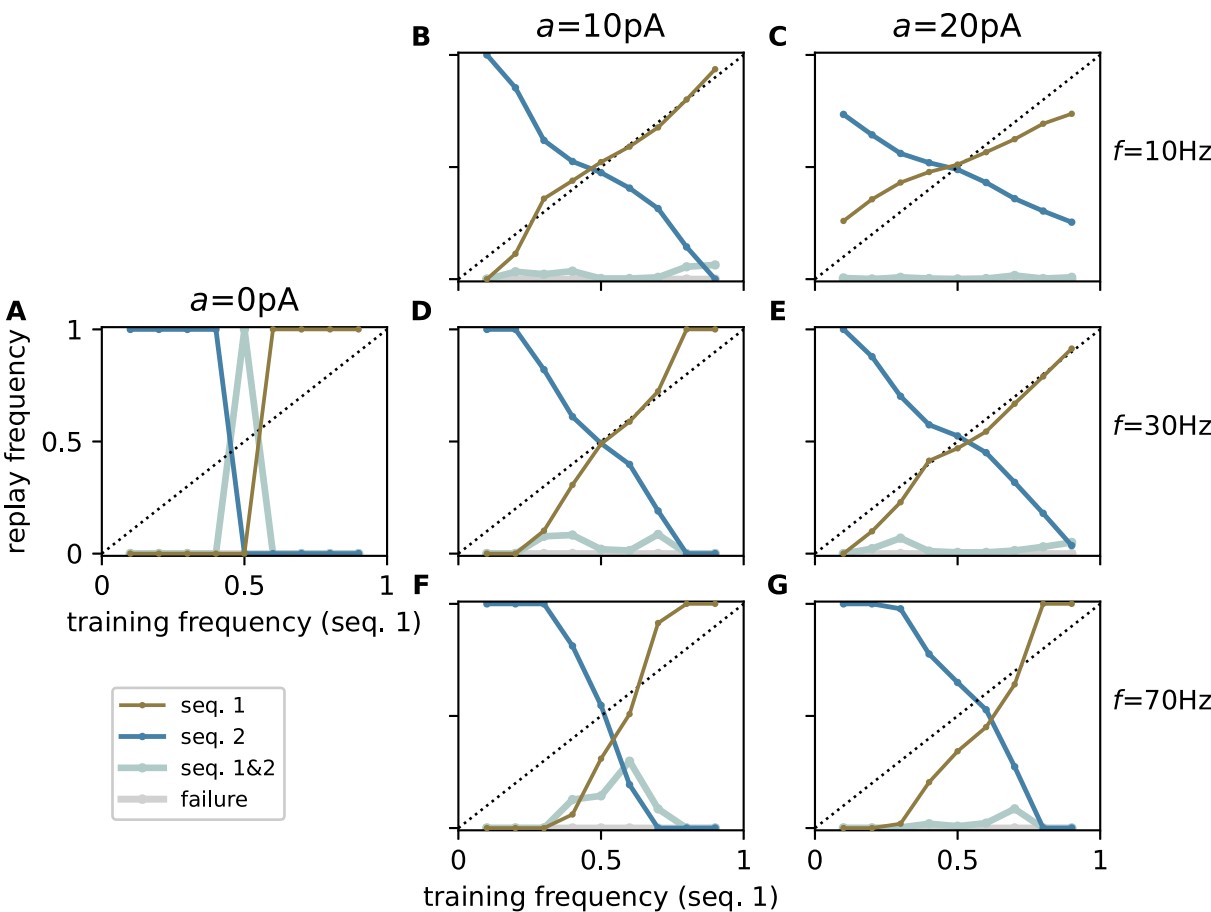

**Fig 9. Changing replay strategy by modulation of spatiotemporal background oscillations.** Dependence of the relative replay frequencies $f_{\{s_1\}}$ and $f_{\{s_2\}}$ of sequences 1 (brown) and 2 (blue), the failure rate $f_\emptyset$ (gray), and the joint probability $f_{\{s_1,s_2\}}$ of replaying both sequences (silver) on the relative training frequency $p_1 = p$ of sequence 1 for different amplitudes $a \in \{0, 10, 20\}$ and frequencies of the background oscillations: $f = 10$ Hz (**B,C**), $f = 30$ Hz (**A,D,E**), and $f = 70$ Hz (**F,G**). Circles represent the mean across $N_t = 181$ trials, averaged across 5 network realizations. See Table 9 for remaining parameters. Same task as described in Fig 2.

strategies by adjusting the noise characteristics, such as the noise or oscillation amplitude, as well as the noise correlation or oscillation frequency.

The sequence processing model proposed here relies on a form of population encoding. In the absence of correlations, noise injected to single neurons therefore largely averages out and leads to a quasi-deterministic and non-exploratory behavior. Locally correlated noise, in contrast, permits an explorative recall behavior where the sequence frequency during learning can be gradually mapped to the recall frequency. Furthermore, noise correlations foster synchronization between neurons within subpopulations, and thereby lead to a more robust context-specific activation of sequences during recall. The problem of noise averaging and the proposed solution are not unique to the model presented here, but are generic for all systems where relevant state variables arise from superpositions of many noisy, uncorrelated components. Fluctuations in the total input current of a single neuron resulting from superpositions of thousands of synaptic inputs, for example, can be efficiently controlled by the level of correlation in the presynaptic activity [47]. Similarly, explorative behavior in other models of population based probabilistic computing [15] can be enhanced by equipping neurons within each population with correlated noise.

Correlation in neuronal firing can originate from both anatomical constraints or network dynamics [23, 24]. In this study, we investigate both types. The first type of noise is implemented in the form of irregular synaptic background input [48–51], where the correlation between neurons of the same subpopulation is resulting from shared presynaptic sources [20, 52]. From an anatomical perspective, this is reasonable as neighboring neurons indeed receive a considerable amount of inputs from identical presynaptic neurons. However, we show that the level of shared-input correlation required for an effective avoidance of noise averaging and maintenance of near synchronous activity is rather high, which contradicts anatomical studies reporting small connection probabilities in local cortical circuits, and hence, low levels of shared input correlation [36–39]. We therefore propose a second, biologically more plausible type of coherent noise resulting from a random stimulus locking to an intrinsic spatiotemporal coherent activity pattern on a large spatial scale, such as waves of cortical activity. Coherent spatiotemporal activity patterns in the cortex are observed in many different forms and under various conditions, including different sleep states, but also in awake behaving animals [27, 42, 45, 46]. Cortical waves can occur spontaneously without being locked to external cues [44]. It is therefore reasonable to assume that the onset time of an external cue is random with respect to the internal state. As shown in this study, this randomness constitutes a natural, locally coherent form of across-trial variability suitable to equip neuronal networks with exploratory behavior. As shown in [44], the timing and position of spontaneous cortical waves before stimulus onset are predictive of the stimulus evoked response and the target detection performance. This is consistent with the model proposed here: the phase of the background oscillation during cue presentation determines the decision outcome. During active vision, cortical waves in the visual cortex have been observed to be tightly locked to the saccade onset [43] and to continue into successive fixation periods [53]. The visual cue, i.e., the fixation onset, is therefore locked to this saccade-triggered oscillating background activity. The eye-movement related modulation of neuronal excitability may hence constitute a mechanism to suppress across-trial variability and lead to more stereotype and reliable responses [44, 54].

In this study, we employ ongoing activity waves as a specific form of coherent spatiotemporal activity, and show that explorative behavior is generated for a range of plausible oscillation frequencies. We propose that a similar behavior can be achieved for other non-oscillatory forms of coherent activity, such as transient propagating wave fronts or bumps [55–57], as well as by other factors modulating the excitability of neighboring neurons in a coherent manner, such as transient neuromodulatory signals. The use of ongoing oscillatory background activity with constant frequency and phase differences is a simplification of this study. A more realistic scenario would be one where each oscillation episode lasts for only few tens or hundreds of milliseconds, and is followed by a new pattern with different phase characteristics. This, however, would not lead to a qualitatively new type of replay behavior as long as two characteristics are preserved: first, at the time of the stimulus arrival, neurons in the same subpopulation experience the same oscillation phase, while neurons in different subpopulations are exposed to different phases, and second, the cue is presented at a different oscillation phase in each trial.

By changing the noise characteristics (such as the amplitude or frequency of the background activity, or the level of correlation), the model proposed in this study can replay competing sequences according to different strategies. For low levels of noise, the network systematically replays the sequence that occurred most often during learning (max-prob). For higher noise levels, it can match the replay frequency to the occurrence frequency during training (probability matching), or become even more explorative. This offers a potential mechanistic explanation of how animals can adjust their decision strategy based on environmental conditions [7]. In the living brain, the noise properties could be controlled by

neuromodulatory signals or by inputs from other brain areas (e.g., during attention; [58]). Our and many other studies predict that, in cases where the decision strategy is shifted towards exploration, more energy needs to be provided for noise generation. In line with this prediction, the work in [59] shows that explorative behavior is accompanied by an increase in the BOLD signal amplitude in cortical areas associated with decision making.

In this study, we equip the network with noise only during sequence replay, but not during training. From a biological point of view, the assumption of vanishing noise during training is not necessarily implausible: as shown in this study, a random locking of the stimulus to an intrinsic coherent spatiotemporal activity pattern may constitute the main cause of exploratory behavior during sequential memory recall. Activity patterns such as traveling waves, however, are not constantly present in the cortex. They may be suppressed during learning, and only added during memory recall to a task-specific extent. Apart from this, the assumption of noise-free training is not critical: in [5], we have shown that the spiking TM model can successfully learn complex sequences in the presence of low and moderate levels of uncorrelated background noise (see supplementary figures S6 and S7 in [5]). Only for large noise amplitudes, the learning performance is impaired as the WTA dynamics are disrupted. If the noise is locally correlated, this effect is less severe because correlated noise increases the response variability across trials, but keeps the variability across neurons in each subpopulation small. Hence, the WTA dynamics remain functional in each trial.

A number of previous studies suggest that synaptic stochasticity, i.e., the variability in postsynaptic responses including synaptic failure [60], may constitute an efficient source of noise for probabilistic computations in neuronal circuits [17, 61]. The total input to a neuron resulting from large ensembles of synapses, however, is likely to be subject to noise averaging. This is in line with an in-vitro study showing that synaptic stochasticity has only a marginal effect on the variability of postsynaptic responses [62]. Averaging of synaptic noise could only be avoided if the variability of synaptic responses was correlated across synapses. To date, it remains unclear how such correlations could potentially arise. Localized neuromodulatory signals or shared presynaptic spike histories may play a role in this.

The spiking TM model employed in this study can adopt a probability-matching strategy because the plasticity dynamics during learning leads to an approximately linear mapping of the relative sequence frequencies during training to the synaptic weights between neurons representing consecutive sequence elements (Fig 4A). The information about the training frequencies is hence stored in the synaptic weights. In this study, we freeze the synaptic weights and preserve this mapping by deactivating the synaptic plasticity dynamics after learning. The spiking TM model can learn the order of items in sequences for a range of different inter-stimulus intervals, but not the timing or the duration of sequence elements. In the replay mode, sequences are replayed with a constant high speed which is mainly determined by the synaptic and neuronal time constants, irrespective of the sequence speed during training [5]. This behavior is reminiscent of the fast, compressed sequence replay observed in hippocampus and neocortex during sleep [63–67]. For our choice of parameters, the inter-element interval during autonomous replay is about 30 ms, which is smaller than the inter-stimulus interval $\Delta T = 40$ ms during training. With an intact plasticity dynamics during replay, the potentiation of synapses between neurons representing consecutive sequence elements would therefore be substantially stronger than during training, because the spike-timing dependent weight increment increases with decreasing pre-post spike intervals in an exponential manner. As the synaptic weights are limited by a hard upper bound $J_{\max}$ (clipping), they would more easily be driven into saturation, such that the information about the training frequency is lost. As a consequence, competing sequences would be replayed in the presence of correlated noise with similar frequencies, irrespective of the training frequencies ("full exploration"; see S3 Fig). In

the absence of noise or for uncorrelated noise, the network still adopts the max-prob strategy. A modification of the STDP dynamics or a thorough tuning of the plasticity parameters may preserve the probability matching performance, even without disabling the plasticity after learning. Alternatively, the spiking TM model may be extended and equipped with additional mechanisms that enable slow sequence replay or even a learning of the sequence speed [3].

For illustration, we have restricted this study to relatively simple sets of $S = 2$ (Figs 3, 4, 6, 7 and 9) or $S = 5$ sequences (Fig 5) with $C = 4$ elements per sequence and 2 overlapping characters. In [5], we have demonstrated that the spiking TM model can successfully learn larger ensembles (up to 6) of longer sequences (up to 12 elements) with larger overlap (up to 10 elements). A systematic investigation of the spiking TM capacity accounting for the maximum number $S$ and length $C$ of sequences as well as the maximum amount of overlap (history dependence) will be subject of future studies (see also [68]). For a larger number $S$ of competing sequences, probability matching becomes harder because the differences $p_i - p_j$ between the relative training frequencies $p_i$ ($i = 1, \ldots, S$) in general become smaller, a consequence of $0 \leq p_i \leq 1$ and $\sum_{i=1}^{S} p_i = 1$. Similar training frequencies lead to similar synaptic weights during the learning process, and in turn, to similar cue response latencies. It is therefore more likely that the winner-take-all dynamics does not come to a unique decision and leads to the joint replay of multiple competing sequences. For the specific choice of noise parameters $\sigma$ and $c$ used here, the replay frequency approximately match the training frequencies.

The spiking TM model introduced in [5] can learn sequences with repeating elements, provided these elements are not immediately following each other. Learning a sequence {A,B,C, B}, for example, is possible, whereas learning of {A,B,B,C} is not. The plasticity dynamics employed in [5] and in this study prohibits a strengthening of connections between synchronously active neurons, i.e., neurons with the same stimulus preference (belonging to the same subpopulation). If the time difference between a presynaptic and a postsynaptic spike is smaller than $\Delta t_{\min} = 4$ ms, a synapse between these neurons is neither potentiated by STDP nor affected by the homeostatic component (see Eqs (13) and (14) in Table 6:Plasticity). Without this restriction, connections between neurons within a subpopulation would quickly grow, in particular at an early learning stage where all neurons within a subpopulations fire in a non-sparse, synchronous manner. As a consequence, the activation of a subset of neurons within some subpopulation would immediately activate other neurons in the same population, and hence trigger a self-prediction. For a sequence {A,B,B,C}, such a self-prediction is indeed wanted, but only in response to the 2nd element. The 1st and the 3rd element must not lead to a self-prediction. Sequences with immediately repeating characters hence require a modification of the plasticity dynamics to permit the strengthening of connections between neurons corresponding to the same character, and at the same time, suppress an excessive growth of synapses between synchronously active neurons.

In the spiking TM model, postsynaptic currents are described by a current-based (CUBA) model where each presynaptic spike triggers a stereotype current response, irrespective of the postsynaptic membrane potential. Real synaptic (and other ionic) currents are mediated by conductances and are determined by the distance of the membrane potential from the respective reversal potential. In combination with point neuron models, the use of conductance-based (COBA) synapses is however problematic as each synapse would feel the same membrane potential, irrespective of its type. In real neurons, synapses on different parts of the neurons, e.g., different dendritic branches, experience different membrane potentials. In this study, we therefore decide in favor of the CUBA synapse model. The neglect of the voltage dependence of the synaptic current is particularly relevant for inhibitory currents. The activation of current-based inhibitory synapses can arbitrarily hyperpolarize the cell membrane (see

**Table 1. Summary of the network model.** Parameter values are given in Table 9.

| Summary | |
|---|---|
| **Populations** | excitatory neurons ($\mathcal{E}$), inhibitory neurons ($\mathcal{I}$), external spike sources ($\mathcal{X}$), background inputs in the form of Poissonian sources ($\mathcal{Q}_k$ and $\mathcal{V}_k$) or sinusoidal current generators ($\mathcal{G}$). $\mathcal{E}$ composed of $M$ disjoint subpopulations $\mathcal{M}_k$ ($k = 1, \ldots, M$) |
| **Connectivity** | • sparse random connectivity between excitatory neurons (plastic) <br> • local recurrent connectivity between excitatory and inhibitory neurons (static) |
| **Neuron model** | • excitatory neurons: leaky integrate-and-fire (LIF) with nonlinear input integration (dendritic action potentials) <br> • inhibitory neurons: leaky integrate-and-fire (LIF) |
| **Synapse model** | exponential or alpha-shaped postsynaptic currents (PSCs) |
| **Plasticity** | homeostatic spike-timing dependent plasticity in excitatory-to-excitatory connections (during training) |

S6 Fig). With a conductance-based (COBA) synapse model, in contrast, the membrane potential is bounded from below by the Cl⁻ reversal potential which is close to the resting potential. Future studies need to investigate to what extent the inhibition-mediated competition mechanisms employed in this study and in [5] are altered if inhibitory currents are described by a COBA model. Further, in the spiking TM model, inhibition is for simplicity mediated by a single inhibitory neuron with very strong and very fast outgoing connections. Future versions of the model could replace this inhibitory neuron by a recurrently connected network of inhibitory neurons with realistic inhibitory weights and time constants. The inhibitory response would still be very fast due to the fast-tracking property of such networks [69].

Overall, our work ties together concepts from sequence processing and decision making in the face of ambiguity. It demonstrates that locally coherent noise is a potential mechanism underlying exploratory behavior, and shows that a random stimulus locking to coherent background activity such as cortical waves constitutes a natural and efficient form of such noise.

## Materials and methods

In the following, we provide an overview of the task and the training protocol, the network model, and the analysis of the sequence replay statistics. A detailed description of the model and a list of parameter values are provided in Tables 1–8 and Table 9, respectively.

### Learning protocol and task

During learning, the network is continuously exposed to repeated presentations of an ensemble of $S$ sequences $s_i = \{\zeta_{i1}, \zeta_{i2}, \ldots, \zeta_{iC_i}\}$ ($C_i \in \mathbb{N}^+, i \in [1, \ldots, S]$) of ordered discrete items $\zeta_{ij}$.

**Table 2. Description of the populations.** Parameter values are given in Table 9.

| Populations | | |
|---|---|---|
| **Name** | **Elements** | **Size** |
| $\mathcal{E} = \cup_{i=k}^{M} \mathcal{M}_k$ | excitatory (E) neurons | $N_\text{E}$ |
| $\mathcal{I}$ | inhibitory (I) neurons | $N_\text{I}$ |
| $\mathcal{M}_k$ | excitatory neurons in subpopulation $k$, $\mathcal{M}_k \cap \mathcal{M}_l = \emptyset$ ($\forall k \neq l \in [1, M]$) | $n_\text{E}$ |
| $\mathcal{Q}_k$ | excitatory Poisson generators, $\mathcal{Q}_k \cap \mathcal{Q}_l = \emptyset$ ($\forall k \neq l \in [1, M]$) | $n$ |
| $\mathcal{V}_k$ | inhibitory Poisson generators, $\mathcal{V}_k \cap \mathcal{V}_l = \emptyset$ ($\forall k \neq l \in [1, M]$) | $n$ |
| $\mathcal{X} = \{x_1, \ldots, x_M\}$ | external spike sources | $M$ |
| $\mathcal{G} = \{g_1, \ldots, g_M\}$ | sinusoidal current generators | $M$ |

**Table 3. Description of the connectivity.** Parameter values are given in Table 9.

| Source population | Target population | Pattern |
|---|---|---|
| **Connectivity** | | |
| $\mathcal{E}$ | $\mathcal{E}$ | random; fixed in-degrees $K_i = K_{\mathrm{EE}}$, delays $d_{ij} = d_{\mathrm{EE}}$, and synaptic time constants $\tau_{ij} = \tau_{\mathrm{EE}}$, plastic synaptic weights $J_{ij}$ ($\forall i \in \mathcal{E}, \forall j \in \mathcal{E}$; "EE connections") |
| $\mathcal{E}$ | $\mathcal{I}$ | all-to-all; fixed delays $d_{ij} = d_{\mathrm{IE}}$, synaptic time constants $\tau_{ij} = \tau_{\mathrm{IE}}$, and weights $J_{ij} = J_{\mathrm{IE}}$ ($\forall i \in \mathcal{I}, \forall j \in \mathcal{E}$; "IE connections") |
| $\mathcal{I}$ | $\mathcal{E}$ | all-to-all; fixed delays $d_{ij} = d_{\mathrm{EI}}$, synaptic time constants $\tau_{ij} = \tau_{\mathrm{EI}}$, and weights $J_{ij} = J_{\mathrm{EI}}$ ($\forall i \in \mathcal{E}, \forall j \in \mathcal{I}$; "EI connections") |
| $\mathcal{I}$ | $\mathcal{I}$ | none ("II connections") |
| $\mathcal{Q}_k$ | $\mathcal{M}_k$ | random; fixed in-degrees $K_i = K_{\mathrm{EQ}}$, delays $d_{ij} = d_{\mathrm{EQ}}$, synaptic time constants $\tau_{ij} = \tau_{\mathrm{EQ}}$, and weights $J_{ij} \in \{0, J_{\mathrm{EQ}}\}$ ($\forall i \in \mathcal{M}_k, j \in \mathcal{Q}_k, \forall k \in [1, M]$; "EQ connections") |
| $\mathcal{V}_k$ | $\mathcal{M}_k$ | random; fixed in-degrees $K_i = K_{\mathrm{EV}}$, delays $d_{ij} = d_{\mathrm{EV}}$, synaptic time constants $\tau_{ij} = \tau_{\mathrm{EV}}$, and weights $J_{ij} \in \{0, J_{\mathrm{EV}}\}$ ($\forall i \in \mathcal{M}_k, j \in \mathcal{V}_k, \forall k \in [1, M]$; "EV connections") |
| $\mathcal{X}_k = x_k$ | $\mathcal{M}_k$ | one-to-all; fixed delays $d_{ij} = d_{\mathrm{EX}}$, synaptic time constants $\tau_{ij} = \tau_{\mathrm{EX}}$, and weights $J_{ij} = J_{\mathrm{EX}}$ ($\forall i \in \mathcal{M}_k, j \in \mathcal{X}_k, \forall k \in [1, M]$; "EX connections") |
| $\mathcal{G}_k = g_k$ | $\mathcal{M}_k$ | one-to-all; fixed synaptic weights $J_{ij} = J_{\mathrm{EG}}$ ($\forall i \in \mathcal{M}_k, j \in \mathcal{G}_k, \forall k \in [1, M]$; "EG connections") |
| all | all | no self-connections ("autapses"), no multiple connections ("multapses") |
| – | – | all unmentioned connections $\mathcal{I} \to \mathcal{I}, \mathcal{V}_k \to \mathcal{V}_k, \mathcal{Q}_k \to \mathcal{Q}_k \ldots \mathcal{X}_k \to \mathcal{M}_l$ ($\forall k \neq l$) are absent |

The order of the sequence elements within a given sequence represents the temporal order of the item occurrence. To investigate the sequence recall performance in the presence of ambiguity, we design the sequences such that they overlap in the first two elements $\zeta_1 = \zeta_{i1}$ and $\zeta_2 = \zeta_{i2}$ ($i \in [1, \ldots, S]$).

The training period is subdivided into $N_e$ episodes. Each training episode is composed of $L$ sequences picked from the set $\{s_1, s_2, s_3, \ldots, p_S\}$ of $S$ training sequences with relative frequencies $p_1, p_2, p_3, \ldots, p_S$, respectively, such that $\sum_{i=1}^{S} p_i = 1$. During training, this set of $L$ sequences is presented repetitively ($N_e$ times) with fixed order. Randomizing the sequence order during training doesn't affect the results provided the relative frequencies are preserved (S4 Fig). The total number $p_i L N_e$ of presentations of a specific sequence $s_i$ during training is proportional to the training frequency $p_i$.

**Table 4. Description of the neuron model.** Parameter values are given in Table 9.

| | |
|---|---|
| **Neuron** | |
| Type | leaky integrate-and-fire (LIF) dynamics |
| Description | dynamics of membrane potential $V_i(t)$ and spiking activity $s_i(t)$ of neuron $i$: <br> • emission of the $k$th spike of neuron $i$ at time $t_i^k$ if <br><br> $$V_i(t_i^k) \geq \theta_i \qquad (3)$$ <br><br> with somatic spike threshold $\theta_i$ <br> • spike train: $s_i(t) = \sum_k \delta(t - t_i^k)$ <br> • reset and refractoriness: <br><br> $$V_i(t) = V_r \quad \forall k, \ \forall t \in (t_i^k, t_i^k + \tau_{\mathrm{ref},i}]$$ <br><br> with refractory time $\tau_{\mathrm{ref},i}$ and reset potential $V_r$ <br> • subthreshold dynamics: <br><br> $$\tau_{\mathrm{m},i} \dot{V}_i(t) = -V_i(t) + R_{\mathrm{m},i} I_i(t) \qquad (4)$$ <br><br> with membrane resistance $R_{\mathrm{m},i} = \frac{\tau_{\mathrm{m},i}}{C_{\mathrm{m},i}}$, membrane time constant $\tau_{\mathrm{m},i}$, and total synaptic input current $I_i(t)$ (see Table 5) <br> • excitatory neurons: $\tau_{\mathrm{m},i} = \tau_{\mathrm{m},E}, C_{\mathrm{m},i} = C_m, \theta_i = \theta_E, \tau_{\mathrm{ref},i} = \tau_{\mathrm{ref},E}$ ($\forall i \in \mathcal{E}$) <br> • inhibitory neurons: $\tau_{\mathrm{m},i} = \tau_{\mathrm{m},I}, C_{\mathrm{m},i} = C_m, \theta_i = \theta_I, \tau_{\mathrm{ref},i} = \tau_{\mathrm{ref},I}$ ($\forall i \in \mathcal{I}$) |

**Table 5. Description of the synapse model.** Parameter values are given in Table 9.

| Synapse | |
|---|---|
| **Type** | continuous, exponential, or alpha-shaped postsynaptic currents (PSCs) |
| **Description** | • total synaptic input current |

$$\text{excitatory neurons:} \quad I_i(t) = I_{\text{ED},i}(t) + I_{\text{EX},i}(t) + I_{\text{EI},i}(t), \quad \forall i \in \mathcal{E}$$

$$\text{inhibitory neurons:} \quad I_i(t) = I_{\text{IE},i}(t), \quad \forall i \in \mathcal{I} \tag{5}$$

with dendritic, inhibitory, excitatory, and external input currents $I_{\text{ED},i}(t)$, $I_{\text{EI},i}(t)$, $I_{\text{IE},i}(t)$, $I_{\text{EX},i}(t)$ evolving according to

$$I_{\text{ED},i}(t) = \sum_{j \in \mathcal{E}} (\alpha_{ij} * s_j)(t - d_{ij}) \tag{6}$$

with $\alpha_{ij}(t) = J_{ij} \dfrac{e}{\tau_{\text{ED}}} t e^{-t/\tau_{\text{ED}}} \Theta(t)$ and $\Theta(t) = \begin{cases} 1 & t \geq 0 \\ 0 & \text{else} \end{cases}$

$$\tau_{\text{EI}} \dot{I}_{\text{EI},i} = -I_{\text{EI},i}(t) + \sum_{j \in \mathcal{I}} J_{ij} s_j(t - d_{ij}) \tag{7}$$

$$\tau_{\text{IE}} \dot{I}_{\text{IE},i} = -I_{\text{IE},i}(t) + \sum_{j \in \mathcal{E}} J_{ij} s_j(t - d_{ij}) \tag{8}$$

$$I_{\text{EX},i}(t) = I_{\text{S},i}(t) + I_{\text{B},i}(t) \tag{9}$$

where $I_{\text{S},i}(t)$ and $I_{\text{B},i}(t)$ are the stimulus and the background input, respectively (see Table 7:Input).

• suprathreshold dynamics of dendritic currents (dAP generation):

• emission of $k$th dAP of neuron $i$ at time $t_{\text{dAP},i}^k$ if $I_{\text{ED},i}(t_{\text{dAP},i}^k) \geq \theta_{\text{dAP}}$

• dAP current plateau:

$$I_{\text{ED},i}(t) = I_{\text{dAP}} \quad \forall k, \quad \forall t \in (t_{\text{dAP},i}^k, t_{\text{dAP},i}^k + \tau_{\text{dAP}}) \tag{10}$$

with dAP current plateau amplitude $I_{\text{dAP}}$, dAP current duration $\tau_{\text{dAP}}$, and dAP activation threshold $\theta_{\text{dAP}}$

• reset: $I_{\text{ED},i}(t_{\text{dAP},i}^k + \tau_{\text{dAP}}) = 0 \ (\forall k)$

• reset and refractoriness in response to emission of $l$th somatic spike of neuron $i$ at time $t_i^l$:

$$I_{\text{ED},i}(t) = 0 \quad \forall l, \quad \forall t \in (t_i^l, t_i^l + \tau_{\text{ref},i}) \tag{11}$$

• reset of $I_{\text{ED},i}$ in case of a strong inhibitory current:

$$I_{\text{ED},i}(t_i^k) = 0, \text{ if } I_{\text{EI},i}(t_i^k) < I_\theta, \tag{12}$$

where $I_\theta$ is the reset dAP current.

After successful learning, the presentation of some sequence element leads to a context dependent prediction of the subsequent stimulus. In case the prediction is wrong the network generates a mismatch signal [5]. As the learned sequence overlap in the first two elements, choosing the cue to be the first sequence element ($\zeta_1$) results in an ambiguity. Here, we investigate the replay frequency of a given sequence $s_i$ as a function of its training frequency $p_i$ and study whether the network can choose between different replay strategies (see Fig 2 and main text).

## Network model

**Network structure.** The network consists of a population $\mathcal{E}$ of $N_E$ excitatory ("E") neurons and a single inhibitory ("I") neuron. The neurons in $\mathcal{E}$ are randomly and recurrently connected, such that each neuron in $\mathcal{E}$ receives $K_{\text{EE}}$ excitatory inputs from other neurons in $\mathcal{E}$. Excitatory neurons are recurrently connected to the single inhibitory neuron. The excitatory population $\mathcal{E}$ is subdivided into $M$ non-overlapping subpopulations $\mathcal{M}_1, \ldots, \mathcal{M}_M$, each of them containing neurons with identical stimulus preference ("receptive field"). Each subpopulation $\mathcal{M}_k$ thereby represents a specific element within a sequence.

**Table 6. Description of the plasticity model.** Parameter values are given in Table 9.

| Plasticity | |
|---|---|
| **Type** | spike-timing dependent plasticity and dAP-rate homeostasis |
| **EE synapses** | • dynamics of synaptic weight $J_{ij}(t)$ (EE connections) during learning:<br><br>$$\forall J_{\min} < J_{ij} < J_{\max} :$$<br><br>$$J_{\max}^{-1} \frac{dJ_{ij}}{dt} = \lambda_+ \sum_{\{t_i^*\}'} x_j(t)\delta(t - [t_i^* + d_{EE}])I(t_i^*, \Delta t_{\min}, \Delta t_{\max})$$<br><br>$$- \lambda_- y_i \sum_{\{t_j^*\}} \delta(t - t_j^*)$$ (13)<br><br>$$+ \lambda_h \sum_{\{t_i^*\}'} (z^* - z_i(t))\delta(t - t_i^*)I(t_i^*, \Delta t_{\min}, \Delta t_{\max}).$$<br><br>$$\forall\{t | J_{ij}(t) < J_{\min}\} : \quad J_{ij}(t) = J_{\min}$$<br><br>$$\forall\{t | J_{ij}(t) > J_{\max}\} : \quad J_{ij}(t) = J_{\max}$$<br>with<br>• list of presynaptic spike times $\{t_j^*\}$,<br>• list of postsynaptic spike times $\{t_i^*\}' = \{t_i^* | \forall t_j^* : t_i^* - t_j^* + d_{EE} \geq \Delta t_{\min}\}$<br>• indicator function<br><br>$$I(t_i^*, \Delta t_{\min}, \Delta t_{\max}) = R(t_i^* - t_j^+ + d_{EE})$$<br><br>$$\text{with} \quad R(\tau) = \begin{cases} 1 & \Delta t_{\min} < \tau < \Delta t_{\max} \\ 0 & \text{else,} \end{cases}$$ (14)<br><br>• maximum weight $J_{\max}$, minimum weight $J_{\max}$, potentiation and depression rates $\lambda_+$, $\lambda_-$, homeostasis rate $\lambda_h$, delay $d_{EE}$, depression decrement $y_i$, minimum $\Delta t_{\min}$ and maximum $\Delta t_{\max}$ time lags between pairs of pre- and postsynaptic spikes at which synapses are potentiated, nearest presynaptic spike time $t_j^+$ preceding $t_i^*$,<br>• spike trace of postsynaptic neuron $i$, evolving according to<br><br>$$\frac{dx_j}{dt} = -\tau_+^{-1} x_j(t) + \sum_{t_j^*} \delta(t - t_j^*)$$<br><br>with presynaptic spike times $t_j^*$ and potentiation time constant $\tau_+$,<br>• dAP trace of postsynaptic neuron $i$, evolving according to<br><br>$$\frac{dz_i}{dt} = -\tau_h^{-1} z_i(t) + \sum_k \delta(t - t_{dAP,i}^k)$$<br><br>with onset time $t_{dAP,i}^k$ of the $k$th dAP, homeostasis time constant $\tau_h$, and<br>• target dAP activity $z^*$ |
| **all other synapses** | non-plastic |

**External inputs during learning.** The network is driven by an ensemble $\mathcal{X} = \{x_1, \ldots, x_{N_{\text{stim}}}\}$ of $M$ external inputs. Each of these external inputs $x_k$ represents a specific sequence element ("A", "B", ...), and feeds all neurons in the subpopulation $\mathcal{M}_k$ that have the same stimulus preference. The occurrence of a specific sequence element $\zeta_{i,j}$ at time $t_{i,j}$ is modeled by a single spike $x_k(t) = \delta(t - t_{i,j})$ generated by the corresponding external source $x_k$.

During training, subsequent sequence elements $\zeta_{i,j}$ and $\zeta_{i,j+1}$ within a sequence $s_i$ are presented with an inter-stimulus interval $\Delta T = t_{i,j+1} - t_{i,j}$. Subsequent sequences $s_i$ and $s_{i+1}$ are separated in time by an inter-sequence time interval $\Delta T_{\text{seq}} = t_{i+1,1} - t_{i,C_i}$.

**Table 7. Description of the input and the output.** Parameter values are given in Table 9.

| Input |
|---|

- prediction mode
  - stimulus
    * repetitive stimulation of the network using the same set $\mathcal{S} = \{s_1, \ldots, s_S\}$ of sequences $s_i = \{\zeta_{i,1}, \zeta_{i,2}, \ldots, \zeta_{i,C_i}\}$ of ordered discrete items $\zeta_{i,j}$ with number of sequences $S$ and length $C_i$ of $i$th sequence
    * presentation of sequence element $\zeta_{i,j}$ at time $t_{i,j}$ modeled by a single spike $x_k(t) = \delta(t - t_{i,j})$ generated by the corresponding external source $x_k$
    * generated current as a response to the presentation of the sequence elements:

$$\tau_S \dot{I}_{S,i} = -I_{S,i}(t) + \sum_{j \in \mathcal{X}} J_{i,j} x_j(t - d_{ij}) \tag{15}$$

    * inter-stimulus interval $\Delta T = t_{i,j+1} - t_{i,j}$ between subsequent sequence elements $\zeta_{i,j}$ and $\zeta_{i,j+1}$ within a sequence $s_i$
    * inter-sequence time interval $\Delta T_{\text{seq}} = t_{i+1,1} - t_{i,C_i}$ between subsequent sequences $s_i$ and $s_{i+1}$
    * example sequence sets:
      - sequence set I: $\mathcal{S} = \{\{A, F, B, D\}, \{A, F, C, E\}\}$
      - sequence set II: $\mathcal{S} = \{\{A, F, B, D\}, \{A, F, C, E\}, \{A, F, G, H\}, \{A, F, I, J\}, \{A, F, K, L\}\}$
- replay mode
  - stimulus
    * presentation of a cue encoding for first sequence elements $\zeta_1$ at time $t^j$, where $j$ denotes the trial number ($j \in [1, \ldots, N_t]$).
    * inter-trial interval $\Delta T_{\text{cue}} = t^{j+1} - t^j$
  - background input in the form of
    * stationary correlated inputs

$$\tau_B \dot{I}_{B,i}(t) = -I_{B,i}(t) + \sum_{j \in \mathcal{Q}} J_{i,j} s_j(t - d) + \sum_{j \in \mathcal{V}} J_{i,j} s_j(t - d) \tag{16}$$

      with Poissonian spike trains $s_j(t)$ of rate $v$, synaptic weight $J_{i,j} \in \{0, J\}$ where $J = J_{\text{EQ}} = -J_{\text{EV}}$, synaptic time constant $\tau_B = \tau_{\text{EQ}} = \tau_{\text{EV}}$, and delay $d = d_{\text{EQ}} = d_{\text{EV}}$
      - variance of $I_{B,i}(t)$ across time:

$$\sigma^2 = \text{Var}(I_{B,i}(t)) = J^2 K v \tau_B, \tag{17}$$

where $K = K_{\text{EQ}} = K_{\text{EV}}$ is the number of either excitatory or inhibitory Poissonian input per excitatory neuron
      - correlation coefficient of $I_{B,i}(t)$ and $I_{B,j}(t)$ across time:

$$c = \frac{\text{Cov}(I_{B,i}(t), I_{B,j}(t))(t)}{\sigma^2} = \begin{cases} 0 & \forall i \in \mathcal{M}_k, \forall j \in \mathcal{M}_l \ (\forall k \neq l) \\ \dfrac{K}{n} & \forall i \in \mathcal{M}_k, \forall j \in \mathcal{M}_l \ (\forall k = l), \end{cases} \tag{18}$$

where $n$ is the number of either excitatory or inhibitory Poissonian sources (see Table 3)
    * or oscillatory currents

$$I_{B,i}(t) = J_{\text{EG}} \cdot a \cdot \sin(2\pi f t + \phi_i) \tag{19}$$

      with amplitude $a$, frequency $f$, and population specific phase $\phi_i = \phi_k \ (\forall i \in \mathcal{M}_k)$

| Output |
|---|

- somatic spike times $\{t_i^k | \forall i \in \mathcal{E}, k = 1, 2, \ldots\}$
- dendritic currents $I_{\text{ED},i}(t) \ (\forall i \in \mathcal{E})$

**External inputs during replay.** After learning the set of sequences $S$, we present cue signals encoding for first sequence elements $\zeta_{\cdot,1}$ by repetitively activating the corresponding external spike source $x_k$ (see above) at $N_t$ time points $t^1, t^2, \ldots, t^{N_t}$. Subsequent cues are separated by an inter-trial interval $\Delta T_{\text{cue},j} = t^{j+1} - t^j$. In section "A spiking neural network recalls sequences in response to ambiguous cues", $\Delta T_{\text{cue},j}$ is constant and in section "Random stimulus locking to spatiotemporal oscillations as natural form of noise", $\Delta T_{\text{cue},j}$ is randomly and uniformly distributed between $u_{\min}$ and $u_{\max}$.

During the replay, excitatory neurons are additionally driven by a background input implemented either in the form of asynchronous irregular synaptic bombardment (see "A spiking neural network recalls sequences in response to ambiguous cues") or oscillatory inputs (see "Random stimulus locking to spatiotemporal oscillations as natural form of noise"). The first

**Table 8. Description of the initial conditions and simulation details.** Parameter values are given in Table 9.

| Initial conditions and network realizations |
| --- |
| • membrane potentials: $V_i(0) = V_\mathrm{r}$ ($\forall i \in \mathcal{E} \cup \mathcal{I}$) |
| • dendritic currents: $I_{\mathrm{ED},i}(0) = 0$ ($\forall i \in \mathcal{E}$) |
| • external currents: $I_{\mathrm{S},i}(0) = 0$ and $I_{\mathrm{B},i}(0) = 0$ ($\forall i \in \mathcal{E}$) |
| • inhibitory currents: $I_{\mathrm{EI},i}(0) = 0$ ($\forall i \in \mathcal{E}$) |
| • excitatory currents: $I_{\mathrm{IE},i}(0) = 0$ ($\forall i \in \mathcal{I}$) |
| • synaptic weights: $J_{ij}(0) \sim \mathcal{U}(J_{0,\min}, J_{0,\max})$ (uniform distribution; $\forall i, j \in \mathcal{E}$) |
| • spike traces: $x_i(0) = 0$ ($\forall i \in \mathcal{E}$) |
| • dAP traces: $z_i(0) = 0$ ($\forall i \in \mathcal{E}$) |
| • connectivity and initial weights are randomly and independently drawn for each network realization |
| **Simulation details** |
| • network simulations performed in NEST [78] version 3.0 [79] |
| • definition of excitatory neuron model using NESTML [80, 81] |
| • synchronous update using exact integration of system dynamics on discrete-time grid with step size $\Delta t$ [82] |
| • source code underlying this study: https://doi.org/10.5281/zenodo.6378376 |

is realized using ensembles of excitatory and inhibitory spike sources $\mathcal{Q}_k$ and $\mathcal{V}_k$ ($k \in [1, \ldots, M]$), each composed of $n$ elements. Each source is an independent realization of a Poisson point process with a rate $\nu$. Excitatory neurons in the same subpopulation $\mathcal{M}_k$ receive $K_{\mathrm{EQ}}$ inputs with weight $J_{\mathrm{EQ}}$ from the ensemble $\mathcal{Q}_k$ and $K_{\mathrm{EV}}$ inputs with weights $J_{\mathrm{EV}} = -J_{\mathrm{EQ}}$ from the ensemble $\mathcal{V}_k$. Spikes from $\mathcal{Q}_k$ and $\mathcal{V}_k$ give rise to a jump in the synaptic current of the postsynaptic cell followed by an exponential decay with a time constant $\tau_{\mathrm{EQ}}$ and $\tau_{\mathrm{EV}} = \tau_{\mathrm{EQ}}$, respectively. The time average input current of a neuron $i$ is

$$\mu_i = 0 \tag{20}$$

and the variance across time

$$\sigma_i^2 = KJ^2 \nu \tau_{\mathrm{B}} \tag{21}$$

where $J = J_{\mathrm{EQ}} = -J_{\mathrm{EV}}$, $\tau_{\mathrm{B}} = \tau_{\mathrm{EQ}} = \tau_{\mathrm{EV}}$, and $K = K_{\mathrm{EQ}} = K_{\mathrm{EV}}$. Given that the populations of background sources are of a finite size, there is a probability that two neurons in the same subpopulation pick a certain number of identical sources, this gives rise to the so called shared input correlation. The correlation coefficient in the input current is governed by

$$c = \frac{K}{n}. \tag{22}$$

With this relationship, we can now vary the correlation coefficient by fixing $K$ and varying $n$. For the special case where $c$ is zero, we assume that each neuron has its own set of independent Poissonian sources. The second type of background input is implemented using an ensemble $\mathcal{G}$ of $M$ sinusoidal current generators $g_k$, each with a frequency $f$, amplitude $a$, and a phase $\phi_k$ ($k \in [1, \ldots, M]$). Excitatory neurons in the same subpopulation $M_k$ receive oscillatory inputs from the same source $g_k$.

Note that the additional background noise described above is not present during the training.

**Neuron and synapse model.**   For all types of neurons, the temporal evolution of the membrane potential is given by the leaky integrate-and-fire model Eq (4). The total synaptic input current of excitatory neurons is composed of currents in distal dendritic branches, inhibitory

**Table 9. Model and simulation parameters.** Parameters derived from other parameters are marked in gray. Curly brackets depict a set of values corresponding to different experiments. Bold numbers depict default values.

| Name | Value | Description |
|------|-------|-------------|
| **Network** | | |
| $N_E$ | {**900**, 1800} | total number of excitatory neurons |
| $N_I$ | 1 | total number of inhibitory neurons |
| $M$ | {**6**, 12} | number of excitatory subpopulations (= number of external spike sources) |
| $n_E$ | $N_E/M = 150$ | number of excitatory neurons per subpopulation |
| $\rho$ | 20 | (target) number of active neurons per subpopulation after learning = minimal number of coincident excitatory inputs required to trigger a spike in postsynaptic inhibitory neurons |
| $n$ | {100, . . ., 1000} | number of excitatory or inhibitory Poissonian sources |
| **Connectivity** | | |
| $K_{EE}$ | {**180**, 360} | number of excitatory inputs per excitatory neuron (EE in-degree) |
| $p$ | $K_{EE}/N_E = 0.2$ | connection probability |
| $K_{EI}$ | $N_I = 1$ | number of inhibitory inputs per excitatory neuron (EI in-degree) |
| $K_{IE}$ | $N_E$ | number of excitatory inputs per inhibitory neuron (IE in-degree) |
| $K_{II}$ | 0 | number of inhibitory inputs per inhibitory neuron (II in-degree) |
| $K_{EQ}$ | 100 | number of excitatory Poissonian inputs per excitatory neuron (EQ) |
| $K_{EV}$ | $K_{EQ} = 100$ | number of inhibitory Poissonian inputs per excitatory neuron (EV) |
| **Excitatory neurons** | | |
| $\tau_{m,E}$ | 10 ms | membrane time constant |
| $\tau_{ref,E}$ | 20 ms | absolute refractory period |
| $C_m$ | 250 pF | membrane capacity |
| $V_r$ | 0 mV | reset potential |
| $\theta_E$ | 20 mV (training), 7 mV (replay) | somatic spike threshold |
| $I_{dAP}$ | 200 pA | dAP current plateau amplitude |
| $\tau_{dAP}$ | 60 ms | dAP duration |
| $\theta_{dAP}$ | 59 pA | dAP threshold |
| $I_\theta$ | −1000 pA | reset dAP current |
| **Inhibitory neurons** | | |
| $\tau_{m,I}$ | 5 ms | membrane time constant |
| $\tau_{ref,I}$ | 2 ms | absolute refractory period |
| $C_m$ | 250 pF | membrane capacity |
| $V_r$ | 0 mV | reset potential |
| $\theta_I$ | 15 mV | spike threshold |
| **Synapse** | | |
| $J_{IE}$ | ∼ 532.76 pA | weight of IE connections (EPSC amplitude) |
| $J_{EI}$ | ∼ −12915.49 pA | weight of EI connections (IPSC amplitude) |
| $J_{EX}$ | ∼ 4112.20 pA | weight of EX connections (EPSC amplitude) |
| $J_{EQ}$ | $\sigma/\sqrt{K_{EQ}\nu\tau_{EQ}}$ | weight of EQ connections (EPSC amplitude) |
| $J_{EV}$ | $-J_{EQ}$ | weight of EV connections (EPSC amplitude) |
| $J_{EG}$ | 1 pA | weight of EG connections (EPSC amplitude) |
| $\tau_{EE}$ | 30 ms | synaptic time constant of EE connections |
| $\tau_{EI}$ | 1 ms | synaptic time constant of EI connections |
| $\tau_{EX}$ | 2 ms | synaptic time constant of EX connection |
| $\tau_{IE}$ | 0.5 ms | synaptic time constant of IE connections |
| $\tau_{EQ}$ | 2 ms | synaptic time constant of EQ connections |
| $\tau_{EV}$ | $\tau_{EQ} = 2$ ms | synaptic time constant of EV connections |
| $d_{EE}$ | 2 ms | delay of EE connections (dendritic) |

*(Continued)*

**Table 9.** (Continued)

| Name | Value | Description |
|---|---|---|
| $d_{\mathrm{IE}}$ | 0.1 ms | delay of IE connections |
| $d_{\mathrm{EI}}$ | 0.1 ms | delay of EI connections |
| $d_{\mathrm{EX}}$ | 0.1 ms | delay of EX connections |
| $d_{\mathrm{EQ}}$ | 0.1 ms | delay of EQ connections |
| $d_{\mathrm{EV}}$ | $d_{\mathrm{EQ}} = 0.1$ ms | delay of EV connections |
| **Plasticity** | | |
| $\lambda_+$ | 0.0009 | potentiation rate |
| $\lambda_-$ | 0.000014 | depression rate |
| $\lambda_{\mathrm{h}}$ | 0.0008 | homeostasis rate |
| $J_{\min}$ | 0 pA | minimum weight |
| $J_{\max}$ | 35 pA | maximum weight |
| $J_{0,\min}$ | 0 pA | minimal initial weight |
| $J_{0,\max}$ | 1 pA | maximal initial weight |
| $\tau_+$ | 20 ms | potentiation time constant |
| $z^*$ | 10.35 | target dAP activity |
| $\tau_{\mathrm{h}}$ | 2200 ms | homeostasis time constant |
| $y_{\mathrm{i}}$ | 1 | depression decrement |
| $\Delta t_{\min}$ | 4 ms | minimum time lag between pairs of pre- and postsynaptic spikes at which synapses are potentiated |
| $\Delta t_{\max}$ | 50 ms | maximum time lag between pairs of pre- and postsynaptic spikes at which synapses are potentiated |
| **Input** | | |
| $S$ | {**2**, 5} | number of sequences per set |
| $C$ | 4 | number of characters per sequence |
| $A$ | {**6**, 12} | alphabet length |
| $N_{\mathrm{e}}$ | {**151**, 101} | number of training episodes |
| $L$ | 10 | number of sequences in a training episode |
| $\Delta T$ | 40 ms | inter-stimulus interval (during training) |
| $\Delta T_{\mathrm{seq}}$ | 100 ms | inter-sequence interval (during training) |
| $N_{\mathrm{t}}$ | 151 | number of cue presentations (trials) |
| $\Delta T_{\mathrm{cue}}$ | 200 ms or $\sim \mathcal{U}(u_{\min}, u_{\max})$ | inter-cue interval |
| $u_{\min}$ | 200 ms | minimal inter-cue interval |
| $u_{\max}$ | 400 ms | maximal inter-cue interval |
| $\sigma$ | {0, 26, 104} pA | noise amplitude resulting from the Poissonian background inputs |
| $c$ | $n/K_{\mathrm{EQ}} \in [0, 1]$ | noise correlation |
| $\nu$ | 1000s$^{-1}$ | rate of Poissonian background inputs |
| $a$ | {0, 10, 20} | amplitude of the sinusoidal current generators |
| $f$ | {10, 30, 70} Hz | frequency of the sinusoidal current generators |
| **Simulation** | | |
| $\Delta t$ | 0.1 ms | time resolution |

currents, and currents from external sources. The inhibitory neuron receives only inputs from excitatory neurons. Individual spikes arriving at dendritic branches evoke alpha-shaped post-synaptic currents, see Eq (6). The dendritic current includes an additional nonlinearity describing the generation of dendritic action potentials (dAPs; NMDA spikes): if the dendritic current $I_{\mathrm{ED}}$ exceeds a threshold $\theta_{\mathrm{dAP}}$, it is instantly set to the dAP plateau current $I_{\mathrm{dAP}}$, and clamped to this value for a period of duration $\tau_{\mathrm{dAP}}$, see Eq (10). This plateau current leads to a long lasting depolarization of the soma. The dendritic input current $I_{\mathrm{ED}}$ constitutes a

simplified, phenomenological description of the effect of NMDA spikes on the somatic membrane potential [70–72]. Similar models have been introduced in previous theoretical studies [73, 74]. For simplicity, we equip each excitatory neuron with only a single dendritic branch, i.e., a single dendritic input current $I_{ED}$. We employ alpha-function shaped postsynaptic dendritic currents with finite rise times to ensure that the response latencies during cue-triggered sequence replay depend on the synaptic weights of connections between excitatory neurons, and hence, on the occurrence frequencies of the learned sequences during training (see section "A spiking neural network recalls sequences in response to ambiguous cues"). Inhibitory inputs to excitatory neurons as well as excitatory inputs to the inhibitory neuron trigger exponential postsynaptic currents, see Eqs (7) and (8). The weights $J_{IE}$ of excitatory synapses on the inhibitory neuron are chosen such that the collective firing of a subset of $\rho$ excitatory neurons in the corresponding subpopulation causes the inhibitory neuron to fire. The weights $J_{EI}$ of inhibitory synapses on excitatory neurons are strong such that each inhibitory spike prevents all excitatory neurons in the network from firing within a time interval of few milliseconds. External inputs are composed of currents resulting from the presentation of the sequence elements or currents from background inputs (see Inputs in Table 7). All synaptic time constants, delays, and weights are connection-type specific.

**Plasticity.** Only excitatory to excitatory (EE) synapses are plastic. All other connections are static. The dynamics of the EE synaptic weights $J_{ij}$ evolve according to a combination of an additive spike-timing-dependent plasticity (STDP) rule [75] and a homeostatic component [76, 77]. During the replay mode, the plasticity is disabled and the EE weights are kept constant (see Table 6 for details about the plasticity).

**Network realizations and initial conditions.** For every network realization, the connectivity and the initial weights are drawn randomly and independently. All other parameters are identical for different network realizations. The initial values of all state variables are given in Tables 8 and Table 9.

**Simulation details.** The network simulations are performed in the neural simulator NEST [78] under version 3.0 [79]. The differential equations and state transitions defining the excitatory neuron dynamics are expressed in the domain specific language NESTML [80, 81] which generates the required C++ code for the dynamic loading into NEST. Network states are synchronously updated using exact integration of the system dynamics on a discrete-time grid with step size $\Delta t$ [82]. The full source code for the implementation with a list of other software requirements is available at Zenodo: https://doi.org/10.5281/zenodo.6378376.

## Sequence replay statistics

We define a sequence $s_i$ to be replayed in response to a cue if more than $0.5\rho$ neurons in the subpopulation representing the last element in $s_i$ fire. The parameter $\rho$ corresponds to the minimal number of neurons that is required to trigger the WTA circuit. It therefore represents the minimal number of active neurons in a subpopulation after successful learning. In the absence of noise, the actual number of active neurons in a subpopulation after successful learning is indeed close to $\rho$ (see [5]). In the present study, we find a similar behavior in the presence of correlated noise (see S2 Fig).

Consider the set $\mathcal{S} = \{s_1, s_2, \ldots, s_S\}$ of $S$ sequences learned by the network. Let

$$\mathcal{P} = \{\emptyset, \{s_1\}, \{s_2\}, \ldots, \{s_1, s_2\}, \{s_1, s_3\}, \ldots, \mathcal{S}\}$$

denote the power set of $\mathcal{S}$, i.e., the set of all subsets of $\mathcal{S}$, including the empty set and $\mathcal{S}$ itself. We define the relative replay frequency $f_{\mathcal{P}_k}$ of each subset $\mathcal{P}_k \in \mathcal{P}$ of sequences as the

normalized number of exclusive replays of this subset $\mathcal{P}_k$, such that

$$\sum_{\mathcal{P}_k} f_{\mathcal{P}_k} = 1. \tag{23}$$

For two sequences $s_1$ and $s_2$, for example, we monitor the four different replay frequencies $f_\emptyset$ (no sequence is replayed), $f_{\{s_1\}}$ (only $s_1$ is replayed), $f_{\{s_2\}}$ (only $s_2$ is replayed), and $f_{\{s_1,s_2\}}$ (both $s_1$ and $s_2$ are replayed). In this work, we refer to $f_\emptyset$ as the "failure rate". Simultaneous replay of both sequences ($f_{\{s_1,s_2\}}$) refers to cases where the network fails at coming to a unique decision.

## Supporting information

**S1 Fig. Adjusting level of correlation permits different replay strategies.** Dependence of the relative replay frequencies $f_{\{s_1\}}$ and $f_{\{s_2\}}$ of sequences 1 (**A, B**) and 2 (**C, D**) on the training frequency $p_1 = p$ of sequence 1 for three different correlation levels $c = 0$, $c = 0.8$, and $c = 1$ (**A, C**), and for a range of correlations (**B, D**). Parameters: noise amplitude $\sigma = 15$ pA and inhibitory weight during replay $J_{EI} = -430.51$ pA adjusted only for connections from the inhibitory neuron to the subpopulation F. The replay frequencies are computed as the mean across $N_t = 151$ trials, averaged across 5 different network realizations. See Table 9 for remaining parameters. (EPS)

**S2 Fig. Response sparsity during replay in the presence of correlated noise.** Dependence of the number of active neurons in the subpopulation corresponding to the last element in {A, F, B, D} (brown) and {A, F, C, E} (blue) on the relative training frequency of sequence 1. The dotted gray horizontal line depicts the target number of active neurons per subpopulation after learning. Noise parameters: $\sigma = 26$ pA, $c = 1$. See Table 9 for remaining parameters. (EPS)

**S3 Fig. Sequence replay in the presence of ongoing synaptic plasticity.** Dependence of **A)** the compound weights (PSC amplitudes) $w_{BF}$ (brown) and $w_{CF}$ (blue), **B)** the population averaged response latencies $t^B$ and $t^C$ for subpopulations "B" (brown) and "C" (blue), **C)** the relative replay frequencies $f_{\{s_1\}}$ and $f_{\{s_2\}}$ of sequences 1 (brown) and 2 (blue), the failure rate $f_\emptyset$ (gray) and the joint probability $f_{\{s_1,s_2\}}$ of replaying both sequences (silver) on the training frequency of sequence 1. In panel A, circles and error bars depict the mean and the standard deviation across different network realizations. In pane B, circles and error bars represent the mean and the standard deviation across $N_t = 101$ trials (cue repetitions), averaged across 5 different network realizations. Note that we run the replay for 200 trials but plotted the statistic of only the last 101 trials. In panel C, circles represent the mean across $N_t = 101$ trials, averaged across 5 different network realizations. Noise parameters: $\sigma = 26$ pA, $c = 1$ (right). See Table 9 for remaining parameters. The data depicted here are results from simulations with enabled synaptic plasticity dynamics during replay. For the results shown in Fig 4, in contrast, the plasticity is disabled during replay to preserve the synaptic weight configuration after the training. (EPS)

**S4 Fig. Sequence replay for randomized sequence order during training.** Dependence of the relative replay frequencies of sequences 1 (brown) and 2 (blue), the failure rate (gray) and the joint probability of replaying both sequences (silver) on the training frequency of sequence 1 for three different noise configurations $\sigma = 0$ pA, $c = 0$ (left), $\sigma = 26$ pA, $c = 0$ (middle), and $\sigma = 26$ pA, $c = 1$ (right). Circles represent the mean across $N_t = 151$ trials, averaged across 5 different network realizations. The data depicted here is generated using the same setting as in Fig

[4F and 4G](), but with a randomized order of sequences during the training.
(EPS)

**S5 Fig. Effect of the learning duration on the probability matching performance.** Dependence of the replay frequencies of sequences 1 (brown) and 2 (blue) of sequence set I, the failure rate (gray) and the joint probability of replaying both sequences (silver) on the number of training episodes. Each episode refers to a set of ten sequences, where each sequence is picked from the set $\{s_1, s_2\}$ with relative frequencies $p_1 = 0.2$ (brown dotted horizontal line) and $p_2 = 1 - p_1 = 0.8$ (blue dotted horizontal line), respectively. Noise parameters: $\sigma = 20$ pA, $c = 1$.
(EPS)

**S6 Fig. Spiking activity (top) and membrane potentials (bottom) at the end of the training and during replay. A,C)** During training (left), the network is exposed to repeated presentations of sequence 1 {A, F, B, D} and sequence 2 {A, F, C, E} (sequence set I) with training frequencies $p_1 = 0.4$ and $p_2 = 0.6$, respectively. Here, only the responses to a single presentation of sequence 1 (black triangles in panel A) are shown at the end of the training period (after 20 episodes). **C,D)** Autonomous replay of sequence 1 in response to activation of sequence element "A" (black triangle in panel C). For clarity, panels A and B show only a small fraction of neurons in each population. Traces in panels C and D depict membrane potentials of two neurons in populations "B" (brown) and "C" (blue), participating in sequences 1 and 2, respectively. During replay, neurons are subject to correlated background noise ($\sigma = 26$ pA, $c = 1$). The resulting membrane potential fluctuations are however small and barely visible in panel D, due to the large hyperpolarizations caused by the global inhibitory feedback. Small bars in panels C and D depict somatic spikes (threshold crossings). In panel D, neurons in both populations "B" (brown) and "C" (blue) generate dAPs (predictions) at about 75ms in response to the ambiguous history "A" and "F". The voltage of the neuron in population "B" (brown) reaches the spike threshold $\theta_E$ (dotted line) first, generates a somatic spike (brown bar), and contributes to the inhibitory feedback leading to the fast and strong hyperpolarization of the neuron in population "C" (blue), and all other excitatory neurons in the network (not shown here).
(EPS)

## Acknowledgments

The authors thank Abigail Morrison, Alexander René and Robin Gutzen for valuable discussions on the project.

## Author Contributions

**Conceptualization:** Younes Bouhadjar, Dirk J. Wouters, Markus Diesmann, Tom Tetzlaff.

**Data curation:** Younes Bouhadjar.

**Formal analysis:** Younes Bouhadjar.

**Funding acquisition:** Dirk J. Wouters, Markus Diesmann, Tom Tetzlaff.

**Investigation:** Dirk J. Wouters, Markus Diesmann, Tom Tetzlaff.

**Methodology:** Younes Bouhadjar, Dirk J. Wouters, Markus Diesmann, Tom Tetzlaff.

**Project administration:** Younes Bouhadjar.

**Software:** Younes Bouhadjar.

**Supervision:** Dirk J. Wouters, Tom Tetzlaff.

**Visualization:** Younes Bouhadjar.

**Writing – original draft:** Younes Bouhadjar.

**Writing – review & editing:** Younes Bouhadjar, Dirk J. Wouters, Markus Diesmann, Tom Tetzlaff.

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
