## [Decision Letter · Decision Letter 0]

22 Nov 2022

Dear Mr Bouhadjar,

Thank you very much for submitting your manuscript "Coherent noise enables probabilistic sequence replay in spiking neuronal networks" for consideration at PLOS Computational Biology.

As with all papers reviewed by the journal, your manuscript was reviewed by members of the editorial board and by several independent reviewers. In light of the reviews (below this email), we would like to invite the resubmission of a significantly-revised version that takes into account the reviewers' comments.

We cannot make any decision about publication until we have seen the revised manuscript and your response to the reviewers' comments. Your revised manuscript is also likely to be sent to reviewers for further evaluation.

Sincerely,

Boris S. Gutkin

Academic Editor

PLOS Computational Biology

Samuel Gershman

Section Editor

PLOS Computational Biology

Reviewer's Responses to Questions

**Comments to the Authors:**

Reviewer #1: General:

The submitted manuscript describes in detail how different types of noise influence sequence replay. This is done using a spiking temporal memory network model which has been trained on short input sequences. The input sequences are presented in different frequencies during training and start with the same elements such that it is not possible to determine the sequence from the first inputs. The network is then tested on a recall task where a sequence is activated by a cue and should be completed by the network. Since sequences can start with the same cue it is ambiguous how the sequence should be completed. The authors look at the networks strategy for completing the sequence (exploitation: always recall the sequence that was observed most frequently during training; probabilistic: recall sequences with probability of how often it was seen during training; exploration: recall all possible sequences with equal probability) and how this is influenced by different types of noise. The main message of the paper is that noise can be used to shift an exploiting strategy towards a probabilistic or explorative strategy if the noise is correlated. Uncorrelated noise cancels itself out. They propose two ways this noise could be produced (shared noise input withing neuron subpopulations and global oscillations in the network to which subpopulations lock depending on input timing) and put out some rough ideas of how this may be realized in biology.

Overall, the paper is well written, easy to follow, and well supported by experiments and figures. The main message is rather simplistic and could probably be expressed much more concisely without the detailed mechanisms of the example network that is being used in this manuscript. The authors even note themselves that “The problem of noise averaging and the proposed solution are not unique to the model presented here” so it could be made a bit clearer why this complex model needs to be introduces in order to make the points presented. Generally, I think the paper could be shortened in a few places but overall, it shows the effect of correlated noise in a nicely structured manner on an interesting example.

Specific comments and questions:

-The example setup on which you demonstrate these results is nice and simplistic, which is good. However, I am still a bit curious whether you tested this in a bit more complex scenario where the network learns more than 2-3 sequences which can overlap and will share transition weights in more places than the first two elements. For instance, what if there are a lot of sequences that contain an F to C transition?

-Line 200 with “feeds” do you mean activates?

-Sentences after like 200 are difficult to follow if you are not deep into the topic. The contain a lot of loaded terms. Is it possible to rewrite this a bit simpler or refer to some references here? I know you mention the materials and methods section but if this is needed to understand this paragraph, I would probably just remove it here.

-Figure 2: Does it matter in which order the sequences are presented during training? Why do you present F? Would this produce the same results if you present sequences ABD and ACE?

-Line 243: How many neurons in a subpopulation fire usually? 50% seems like a low threshold. Is it common for many neurons in the subpopulation not to fire? How many neurons can fire in other populations if for instance D is recognized? It would be interesting to have a bit more details or statistics to get an intuition here.

-Figure 3: does this show actual results or is this a hypothetical visualization? You present stimulus A only 3 times, are the outcomes picked to be representative for the general distribution? In the correlated noise example, I would assume it could also happen that the distribution on 3 examples is not 1:2.

-Figure 3 C: Can you elaborate more on why there are fewer red dots (inhibitory activations) in the second row?

-Figure 3 A: Do all arrows to one sub population have the same weight wCF?

-Figure 3 caption: I don’t think you can assume people know what WTA means.

-Lines 247-253 seem a bit repetitive to what has already been said earlier

-Line 255: What determines this if there is no noise?

-Line 263 missing word “that” at end of line

-Line 267: You don’t have noise during learning, do you have a reason why this is a plausible assumption to make? Do you have an idea what effect correlated noise during learning would have?

-2 paragraphs starting from line 270 seem to repeat a lot of information from the beginning (but adding formulas) maybe it can be cut down in places

-Figure 4 caption: What distinguishes the “5 different network realizations”?

-Chapter starting in line 319: The paragraphs jump between biology and your results a bit. After reading the start of the first paragraph I was wondering “can you assume control over noise? Where would this come from?” You allude to that towards the end of the second paragraph. Maybe you could combine the biology. Results after line 329 seem to be a bit repetitive again, maybe that could be cut a bit shorter.

-Line 333: What to you mean with “changing the effective weights”?

-Line 345-345: Do you have a source for that? How does your mechanism work if they become locked after some observations?

-Lines 348-350: This is the same setup you explained already, right?

-Lines 363-364: I’m not sure I understand this, can you maybe elaborate a bit more what you mean here?

-Lines 365-368: A bit confusing to me. It seems like a clear conclusion is missing here.

Reviewer #2: Human and non-human primates can adjust their decision strategy based on the relative frequency of experienced outcomes, e.g. in game tasks. The present manuscript explores the modulation of decision strategies in a stimulus sequence completion task through modulation of the recall mechanism in a spiking neural network model by the level of correlated noise. This study builds on an earlier publication in PLOS CB that had introduced the simple network architecture with N populations of excitatory neurons (E) and a single inhibitory neuron that reciprocally connects to all excitatory neuron. Spike timing dependent plasticity at all E-E synapses allows to learn or ‘imprint’ a sequence of stimulus cues, where each E population is activated by a specific stimulus cue with high temporal precision.

The novelty and strength of the present extension of this work lies in the detailed study of the influence of noise and, alternatively, the phase-locking to oscillations. The authors can show that independent Poisson noise averages out and effectively represents the deterministic case. However, correlated noise or (partially) shared input across neuronal subpopulations can alter the network behavior during sequence recall and in an ambiguous task where different sequences of cues/syllables are initially congruent (first two syllables). During the initial training phase these sequences have been presented with different relative frequencies. The authors show that the amplitude of noise and the degree of noise correlation has a mechanistic effect and explain the emergence of different recall strategies (shown in central figures 4 and 5).

We still lack understanding of the emergence of (or control over) decision strategies in the brain by means of biologically plausible mechanisms. The present study provides highly valuable insight and makes a contribution at both, the conceptual and mechanistic level, showing how switching defined noise characteristics in a simple spiking network model allows to control switching between decision strategies.

This study very well fits the scope of PLOS CB and deserves publication in revised form. Below I provide a number of major review comments to be addressed during revision that are mainly concerned with the description of methods that lack clarity and transparency in several parts, and with the biological realism of the model (including single neuron biophysics) that currently is neither transparently described nor discussed.

Major

#1 Additional questions to the model

Here I formulate a few questions that could possibly be addressed in an extended model analysis to provide more insight in model function and capacity; these are suggested additions and I do not expect or require the authors to address all of them.

(I) What is the capacity in terms of number of sequences to be represented in probability matching? How does performance depend on the number of training runs?

(II) Currently the sequence lengths is four cues from a set of six (corresponding to six populations). What is the capacity of the model in terms of sequence length (more than 4) and performance for sequences where the same cues reoccur, e.g. AABB? Does a larger number of populations (e.g. 12 instead of six) alter any of the conclusions (i.e. can this model be scaled to a larger set of syllables).

(III) How ‘badly’ does the model perform or how much would the results deviate if the STDP rule was not disabled during replay? If it would still perform reasonable, then this would be an interesting result and conceptually better aligned with theories of continual learning.

(IV) What are the necessary conditions under which the network can be trained successfully? Can the inter-stimulus interval within a sequence easily be changed (e.g. to 25ms instead of 50ms) without loss of function? Are any of the neuron / synapse / STDP parameters matched to the 50ms inter-stimulus interval within a sequence? I think this is an important point that should be addressed as generally we would expect sequential task to vary strongly in inter-stimulus duration while network and synapse parameters can be expected to be fixed.

(V) l.267: “During training, the weak noise employed here hardly affects the network behavior as the external inputs (stimulus) are strong and lead to a reliable, immediate responses.“ If the noise level was not altered but kept constant throughout learning and replay, what are the conditions of successful learning with respect to the stimulus? Currently the stimulus is deterministic and reliable enforcing a single spike in each stimulated neuron, this seems a rather unrealistic condition in the brain.

#2 Learning protocol and task

The definition of the learning protocol in l.99ff is poor. How often are sequences presented overall during training, this is neither mentioned here not in the main text? Is this depending on the relative frequencies of the sequences or/and on the number of different sequences (e.g. 2 vs. 3 different sequences) or is always the same total number of training trials applied? What is the performance and strategy for few trial learning (in the order of ~10)?

The description of the task in l.99ff and of the task performance in l.177ff is also unclear. I understand that there are three strategies that can be desirable and thus as low or high performance would be dependent on the strategy. The four basic measures in l.181 are thus rather counting statistics of sequences that subserve derived quantification of task performance?

#3 Changes from the learning to the replay network

The Results text states: “The model can be configured into a replay mode, where the network autonomously replays learned sequences in response to a cue stimulus. This is achieved by changing the excitability of the neurons such that the activation of a dAP alone can cause the neurons to fire” (l.227f). My understanding of the re-configuration from training to replay mode is a considerably more drastic one and this must be made more explicit and transparent!

When switching from learning to replay ‘mode’ the authors device at least FOUR major changes to the network model. (I) The STDP is disabled (what is a biologically plausible mechanism, dependency of STDP on neuromodulatory input? Is there any experimental evidence? How does replay perform if STDP is not disabled?); (II) external noise is now providing background while there was zero noise during learning; (III) The inhibitory synaptic weight is decreased by almost a factor of 6 (now still about 4 times larger than the excitatory weight); (IV) Excitatory weights are reduced by a factor of about 2. How do these changes translate in the statement above on the activation by dAP alone? Did I miss any other modifications?

This switch of operational ‘mode’ and change in basic parameters is a very strong supervised modification of the network and the biophysics of the synapses. This needs to be described transparently in the Method section and it must at least be briefly described in the Result section, possibly also in the caption of Fig. 1 or by means of an additional sketch in Fig. 1. Please discuss the biological plausibility / potential mechanisms for each change that underlies the switch of the network ‘mode’.

#4 Role and plausibility of inhibition

l.160 “each inhibitory spike prevents all excitatory neurons in the network that have not generated a spike yet from firing.“ - the description as „not … yet“ is unclear; is spiking artificially prevented for the rest of the input sequence/trials in all neurons that have not spiked yet? Or is this paraphrasing the likely effect of the extremely strong inhibitory synapse (see below)? Also, why should those excitatory neurons “that have” already spiked still be able to generate a spike, this does not make sense to me. Please describe in clear terms!

The very strong inhibitory synaptic current of about -13 nA (Table 2) is problematic because it seems biologically unrealistic. The fact that the authors use a current-based rather than a conductance-based neuron model now becomes relevant. At this point the current based model does no longer approximate the biophysics of a biological neuron where the reversal potential should prevent such large outward charge fluxes and the expected extreme hyperpolarization. This means that the injection of a current profile with -13nA amplitude, resulting in a gigantic charge transfer, is to be considered as a completely artificial intervention. The deliberate effect is probably the complete shutdown of excitatory spiking for a time period longer than the experiment/trial. It would thus be great and transparent to provide (e.g. in supplements) insight in the single neuron propagation/physiology and show in a figure both, the integrated current and membrane potential for e.g. two excitatory and the single inhibitory neuron as a function of time during at least one single learning sequence and a single replay of that sequence. This would allow additional bottom-up insight into the model function that is currently not provided by the more abstract presentation of the few replay spike trains and the derived measures of latencies and relative replay frequencies.

The fact that there is only a single inhibitory neuron that is to my interpretation connected to all excitatory neurons (“Excitatory neurons are recurrently connected to the single inhibitory neuron“) means a complete suppression of any excitatory output. How could the inhibitory neuron then “meditate competition” (l.198, caption Fig. 3 etc.)? Is this competition only devised during replay (due to 6 times lower synaptic weights?)? This needs to be introduced and explained transparently in the Results and Methods section. Could the authors think of a biologically realistic improvement of their model of inhibition e.g. by introducing a pool of inhibitory neurons, each with a realistic inhibitory effect on the E neurons?

#5 Neuron dynamics

I understand that alle neurons are point neurons and the voltage dynamics is governed by eqn 6 and all input currents simply add up (eqn 7). Therefore, the term “dendritic current” I_ED really is only an interpretation and not reflected in any distal input with length constants or compartments of a dendrite with geometry.

The reasoning behind the properties of I_ED needs to be outlined and the authors should make more explicit that this is why they interpret these inputs as the somatic effect of distal dendritic inputs. Currently the authors make no claims and do not provide references to why the properties of I_ED represent a somatic effect of distal excitatory dendritic inputs. Specifically, why is the PSC alpha-shaped, should this reflect a dendritic filtering and if so, should it not be a beta-shaped current with a second time constant? More important, why is the dendritic AP represented by a constant plateau current (that is seemingly not filtered or damped during its travel to the soma)? Is the biological motivation the current effect of a dendritic Ca spike or a dendritic Na spike or something else? Please clarify in the text. The fact that the threshold is applied to the sum of all dendritic inputs means that the dAP is not dendritic branch specific, a yet unexplored input dimension of the model.

#6 Latency analysis unclear

I was a little confused with the latency analysis. The definition in line 273ff seems to be restricted to a single subscript index “s”. Do the latencies in Fig. 4 show latencies to one particular stimulus in the sequence or the average latency?

In l.273 “(time of first spike after the cue)“ – this is unclear: The absolute numbers of latency in the order of 50ms together with the spike times in Fig. 3B-D indicate that the latency is computed from the spike times relative to the *previous* cue/stimulus and not the present? Please clarify.

Related to this: in l.237ff subscript „s“ is not defined. Does it denote the number or the type of stimulus in a sequence? Ambiguity with sequences denoted as s1, s2, … (l. 177) is possible.

The symbol „t“ is used for timing of stimuli/cues (l.121) and for spike time and for the average spike latency (l.273). What is the difference between subscript and superscript indices i,j in lines 120-124 and lines 127 and 247f, this is confusing. Is superscript type and subscript number of stimulus within a sequence?

Minor

The Ordering of “Materials & Methods” (1st) and “Results” (2nd) sections does not adhere to the guidelines of PLoS CB. At the end of the Introduction the authors point out, that the reader may proceed to the Results section. This is not necessary if the order is reversed and this might refer to an earlier submission elsewhere.

Caption Fig. 1: “two sequences A,F,C,E (brown) and A,F,B,D (blue)“ – colors are exchanged.

l.157: “(9-10)” -> “Eq. (9-10)”

l.157: “The weights J_IE of excitatory synapses on inhibitory neurons..:“ – should be singular “inhibitory neuron”.

Is “cue” and “stimulus” used interchangeable or are they defined differently?

Towards the end of the discussion the authors argue that “The total input to a neuron resulting from large ensembles of synapses, however, is likely to be subject to noise averaging, unless the variability of synaptic responses is correlated across synapses.“ To this end and in line with the authors’ authors argument on averaging of independent (synaptic) noise the following reference provides a combined experimental-theoretical analysis of the effect of synaptic amplitude variability on postsynaptic input summation:

Nawrot, M. P., Schnepel, P., Aertsen, A., & Boucsein, C. (2009). Precisely timed signal transmission in neocortical networks with reliable intermediate-range projections. Frontiers in Neural Circuits, 1.

**Have the authors made all data and (if applicable) computational code underlying the findings in their manuscript fully available?**

Reviewer #1: Yes

Reviewer #2: Yes

PLOS authors have the option to publish the peer review history of their article (what does this mean?). If published, this will include your full peer review and any attached files.

Reviewer #1: **Yes: **Viviane Clay

Reviewer #2: **Yes: **Martin Nawrot
---

## [Decision Letter · Decision Letter 1]

2 Mar 2023

Dear Mr Bouhadjar,

We are pleased to inform you that your manuscript 'Coherent noise enables probabilistic sequence replay in spiking neuronal networks' has been provisionally accepted for publication in PLOS Computational Biology.

Best regards,

Boris S. Gutkin

Academic Editor

PLOS Computational Biology

Samuel Gershman

%CORR_ED_EDITOR_ROLE%

PLOS Computational Biology

Reviewer's Responses to Questions

**Comments to the Authors:**

Reviewer #1: Thank you for your thorough response to my review and the updates to the manuscript. All my concerns and questions have been addressed.

Reviewer #2: The authors have provided detailed answers to all my questions and concerns. During revision the authors have clearly improved the clarity of methods description. They followed a number of my suggestions. Additional points have been discussed. I specifically liked the additional result in supplemantal Fig. S4 (R2:3), the additional result in Fig. S3 that demonstrates a loss of function if STDP remains active during recall (R2:7), the fact that tuning of the PSC time constant reduced complexity such that inhibitory and excitatory time constants are now constant across training and replay phases (R2:13)

I endorse publication of the article. Still, I have a few final comments for the authors that may be consindered before publication:

R2:4 - The figure shown there should it appear in the MS / supplements but I did not find it.

R2:7 - Here the authors state that performance gradually degrades if STDP is active during recall. However, Fig. S3 indicates to me that the performance broke down completely during the 2nd half of the replay.

R2:15 - Here the authors could not fully convince me. I think it would be insightfull to show voltage traces and spikes across time in a supplemental figure during complete sequence presentations in training and replay setting. Pointing to an earlier paper is only partially helpful.

**Have the authors made all data and (if applicable) computational code underlying the findings in their manuscript fully available?**

Reviewer #1: Yes

Reviewer #2: Yes

PLOS authors have the option to publish the peer review history of their article (what does this mean?). If published, this will include your full peer review and any attached files.

Reviewer #1: **Yes: **Viviane Clay

Reviewer #2: No

---

## [Editor Report · Acceptance letter]

22 Mar 2023

PCOMPBIOL-D-22-01079R1 

Coherent noise enables probabilistic sequence replay in spiking neuronal networks

Dear Dr Bouhadjar,

I am pleased to inform you that your manuscript has been formally accepted for publication in PLOS Computational Biology. Your manuscript is now with our production department and you will be notified of the publication date in due course.

With kind regards,

Timea Kemeri-Szekernyes
